# Preconditioned Langevin Dynamics with Score-Based Generative Models for Infinite-Dimensional Linear Bayesian Inverse Problems

**Lorenzo Baldassari**
University of Basel
lorenzo.baldassari@unibas.ch

**Josselin Garnier**
Ecole Polytechnique, IP Paris
josselin.garnier@polytechnique.edu

**Knut Sølna**
University of California Irvine
ksolna@uci.edu

**Maarten V. de Hoop**
Rice University
mvd2@rice.edu

## Abstract

Designing algorithms for solving high-dimensional Bayesian inverse problems directly in infinite-dimensional function spaces—where such problems are naturally formulated—is crucial to ensure stability and convergence as the discretization of the underlying problem is refined. In this paper, we contribute to this line of work by analyzing a widely used sampler for linear inverse problems: Langevin dynamics driven by score-based generative models (SGMs) acting as priors, formulated directly in function space. Building on the theoretical framework for SGMs in Hilbert spaces, we give a rigorous definition of this sampler in the infinite-dimensional setting and derive, for the first time, error estimates that explicitly depend on the approximation error of the score. As a consequence, we obtain sufficient conditions for global convergence in Kullback–Leibler divergence on the underlying function space. Preventing numerical instabilities requires preconditioning of the Langevin algorithm and we prove the existence and the form of an optimal preconditioner. The preconditioner depends on both the score error and the forward operator and guarantees a uniform convergence rate across all posterior modes. Our analysis applies to both Gaussian and a general class of non-Gaussian priors. Finally, we present examples that illustrate and validate our theoretical findings.

## 1 Introduction

Inverse problems arise in many challenging applications, such as X-ray computed tomography, seismic tomography, inverse heat conduction, and inverse scattering. These problems share a common goal: to estimate unknown parameters from noisy observations or measurements [1]. What makes them difficult is that they are often ill-posed in the sense of Hadamard [2]: they may have multiple solutions, no solutions at all, or solutions that are highly sensitive to small perturbations in the data. A possible approach to address these difficulties is to cast the problem in a probabilistic framework known as Bayesian inference. In the Bayesian approach, one first specifies a prior distribution that encodes knowledge about the unknown before any data is observed, along with a model for the observational noise. Bayes' rule is then used to update this prior knowledge in light of the measurements, yielding the so-called *posterior distribution*, which describes the distribution of the unknown conditioned on the data. By sampling from the posterior one can extract statistical information and quantify uncertainty in the solution [3–6].

39th Conference on Neural Information Processing Systems (NeurIPS 2025).

A central challenge in applying Bayesian inference to inverse problems is that in many cases—especially those governed by partial differential equations (PDEs)—the unknowns to be estimated are *functions* that lie in a suitable function space, typically an infinite-dimensional Hilbert space. It is therefore crucial to design Bayesian inference algorithms that are both theoretically sound and computationally effective in arbitrarily high dimensions. A way to achieve this is by lifting these problems to an infinite-dimensional space and designing inference methods directly in that setting. This approach, sometimes referred to as "*apply-algorithm-then-discretize*"—or, in the context of Bayesian inference, "*Bayesianize-then-discretize*"—allows for the development of algorithms that are inherently discretization-invariant, as the Bayes formula and algorithms are properly defined on Hilbert spaces [3, 7]. In contrast, the opposite approach—"*discretize-then-Bayesianize*"—can lead to several issues, such as instability as the discretization of the underlying problem is refined, or worse, methods that seem stable but whose results are theoretically implausible [8, 9].

These considerations manifest clearly even in simple scenarios. In Figure 1 we consider two examples involving a vanilla diffusion Langevin sampler. In the first one, we sample from a Gaussian posterior. While the method appears numerically stable and produces samples with seemingly reasonable behavior, a closer inspection shows that the samples carry infinite energy—they do not belong to the infinite-dimensional Hilbert space. That is, the algorithm is producing objects that are not valid functions in the limit of refined discretization. In the second example, we attempt to fix this by choosing a *trace-class* prior, which ensures that samples have finite energy and are well-defined in a Hilbert space. This theoretically-motivated structure, however, comes at a cost: without adjustments, the drift of the vanilla Langevin sampler may diverge at fine scales.

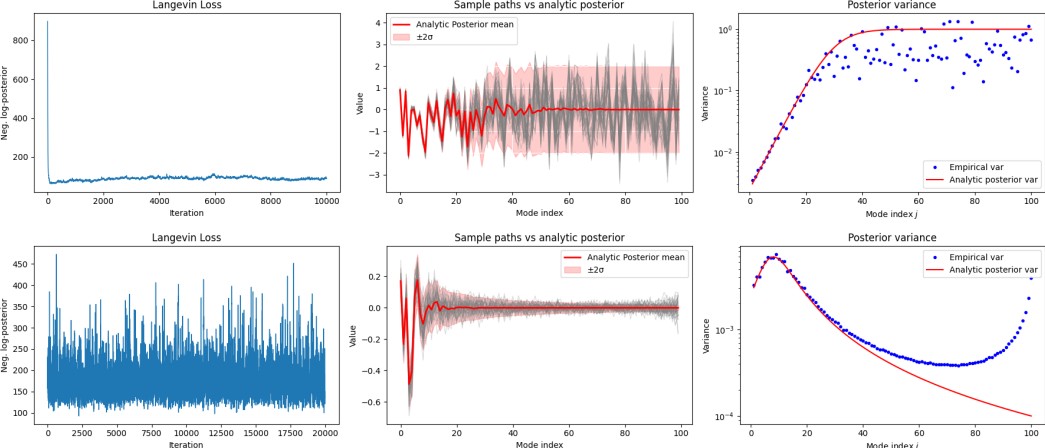

Figure 1: We consider the toy linear inverse problem $y_j = A_{jj}X_0^{(j)} + n_j$ in the basis $(v_j)$ of the Hilbert space $H$, with $A_{jj} = e^{-0.1j}$ and $n_j \sim \mathcal{N}(0, 0.05^2)$, for $j \leq 100$. In the top row, we sample the posterior using an identity prior covariance on $X_0$. The Langevin diffusion seems stable, but the eigenvalues of the posterior covariance do not decay at infinity and therefore the samples do not belong to the Hilbert space $H$. In the bottom row, we use a trace-class covariance prior with diagonal terms $\sim 1/j^2$; the drift of the vanilla Langevin sampler starts diverging at fine scales.

These types of challenges, intrinsic to the infinite-dimensional setting, have long been studied in the Bayesian inverse problems community, but are now receiving renewed attention with the rise of deep learning methods for posterior sampling. One popular class of methods that still lacks a complete theoretical understanding in this context is *score-based generative models* (SGMs), which generate samples from complex distributions by first learning the (Stein) score—the gradient of the log-density [10]—and then using it in various sampling algorithms [11, 12]. For example, [13] employs the learned score in a Langevin-based sampler, while [14] unified SGMs and diffusion-based methods [15, 16] through a stochastic differential equation (SDE) framework, known as score-based diffusion models. After their introduction, SGMs have been applied successfully to Bayesian inverse problems, either by learning the score conditioned on data [17–19], or by using the score of the prior distribution—the *unconditional score model*—within Langevin-type samplers. Crucially, with a few exceptions [20–23], these approaches assume that the posterior is supported in a finite-dimensional space, leaving the challenges of infinite dimensions to heuristics and ad hoc solutions.

In this work, we present a detailed analysis of SGMs for Bayesian inference of linear inverse problems, going beyond the common assumption that the posterior is supported on a finite-dimensional space. We focus on a widely used posterior sampling technique that combines SGMs—used as powerful learned priors to capture complex features—with a Langevin-based sampler [24–26]. Lifting the problem directly to function spaces is not a mere technicality: we show that, to provably sample the posterior, the Langevin diffusion must be modified by a preconditioning operator $C$ acting on the Hilbert space. This preconditioner is not an ad hoc fix but rather built into the fabric of the infinite-dimensional setting: it first appears in the forward diffusion process (5) whose time-reversal is used to learn the prior, and must then be carried through into the Langevin sampler to ensure convergence to the correct posterior (Section 3). Crucially, $C$ cannot be the identity: for the time-reversed diffusion to remain Hilbert-space-valued, $C$ must be trace class. Setting $C = I$ leads to the same theoretical and numerical issues as highlighted in Figure 1 above.

The importance of preconditioning in function spaces has been well established in the context of posterior sampling [3, 8, 27–30], but its implications have not yet fully explored for infinite-dimensional SGMs. In this setting, we characterize the interplay between the preconditioner $C$, the trace-class prior, the score approximation error, and the linear forward map of the inverse problem. In particular, we analyze the impact of the score approximation error at small times—where the score is learned in practice—and identify a *theoretically optimal preconditioner* that ensures uniform convergence rates across posterior modes (Section 4). We carry out the analysis by focusing on two cases: a Gaussian prior measure, and a more general class of priors which are absolutely continuous with respect to a Gaussian measure (Section 5). Illustrations are provided in Section 6.

**Related Work.** There exists a large body of literature on infinite-dimensional MCMC algorithms [3, 8, 27–40], which include a variety of preconditioning strategies for posterior sampling. However, these works precede the recent wave of papers on SGMs and therefore do not address the central focus of our analysis: the interplay between the score approximation error, the preconditioning operator, the trace-class prior, and the sampler convergence, which we study in detail in both the Gaussian and non-Gaussian settings.

Closest to our work are the papers that use SGMs for posterior sampling, such as [21, 24–26], which employ SGMs as learned priors in a Langevin-type diffusion algorithm. Among these works, the theoretical analysis of [25] is the most directly related to ours. However, there are key differences. [25] analyze Langevin dynamics with SGMs for posterior sampling in finite dimensions, as their results provide convergence error estimates that explicitly depend on the score approximation error but diverge as the dimension of the problem increases. In contrast, our error analysis, since it is formulated directly in infinite dimensions, provides conditions to ensure global boundedness (Theorem 3.1). Moreover, the finite-dimensional setting of [25] does not address the role of preconditioning, which becomes essential in infinite dimensions. Other related works include [41, 42] which investigate preconditioning in Langevin dynamics with SGMs. However, these analyses are also finite-dimensional and do not account for the score approximation error. As a result, they do not capture the critical role of preconditioning, which—as we show in Section 4—becomes crucial in function spaces.

As we have pointed out several times, the learned score plays a key role in our analysis. Among the theoretical frameworks defining SGMs in infinite dimensions [23, 43–48], we follow those of [20, 49] for continuous-time diffusions. An important contribution of our work is to show that the convergence bound depends explicitly on the accuracy of the approximated score, and that controlling this error is key to designing a preconditioner that ensures convergence in function spaces (Theorem 4.1).

Finally, we note that [21] also explores the role of preconditioning to ensure convergence in infinite dimensions in the context of SGMs. Their analysis is conducted in a more complex setting—nonlinear inverse problems. Their argument builds on the proof of [25], but the difficulties of the nonlinear setting prevent them from identifying an optimal preconditioner. In contrast, our work takes full advantage of the linear setting, where the distributions at play admit explicit formulas. This allows us to derive detailed error estimates in the small diffusion time regime, where the score is typically learned, and discuss the impact of the score approximation error on posterior sampling—including the effects of preconditioning on the bias error. Furthermore, their analysis focuses only on Gaussian priors, while we generalize and consider non-Gaussian priors (Section 5).

## 2 Langevin Posterior Sampling with Score-Based Generative Priors

We work in the setting of a linear Bayesian inverse problem formulated in infinite dimension. Let $H$ be a separable Hilbert space with inner product $\langle \cdot, \cdot \rangle$, and let $C, C_\mu : H \to H$ be trace-class, positive-definite, symmetric covariance operators. The unknown quantity of interest is an $H$-valued random variable $X_0 \sim \mu$, where the prior measure $\mu$ is assumed to be absolutely continuous with respect to a Gaussian reference measure $\mathcal{N}(0, C_\mu)$, with density

$$\frac{d\mu}{d\mathcal{N}(0, C_\mu)}(X) \propto \exp\big(-\Phi(X)\big). \tag{1}$$

The observations $y \in \mathbb{R}^N$ are modeled as

$$y = AX_0 + n,$$

where $A : H \to \mathbb{R}^N$ is a linear operator, and $n \sim \mathcal{N}(0, \sigma^2 I_N)$ is Gaussian observational noise independent of $X_0$. Since we consider an observational model corresponding to observing a finite-dimensional subspace of $H$, there exists an orthonormal basis $(v_j)$ of $H$ such that $A v_j = 0$ for all $j > N$. Let $(e_j)$ denote the standard basis of $\mathbb{R}^N$. Then the observation model can be written as $y_i = \sum_{j=1}^{N} A_{ij} X_0^{(j)} + n_i$, where $A_{ij} = \langle e_i, A v_j \rangle$, $y_i = \langle y, e_i \rangle$, $X_0^{(j)} = \langle X_0, v_j \rangle$, and $n_i = \langle n, e_i \rangle$.

The posterior distribution $\pi_y$ of $X_0$ conditioned on the observations $y$ is absolutely continuous with respect to $\mathcal{N}(0, C_\mu)$:

$$\frac{d\pi_y}{d\mathcal{N}(0, C_\mu)}(X) \propto \exp\Big(-\Phi(X) - \frac{1}{2\sigma^2}\|AX - y\|^2\Big). \tag{2}$$

The goal of infinite-dimensional Bayesian inference is to design sampling methods for $\pi_y$ whose performance remains stable as the underlying discretization is refined. To this end, we study a widely used sampler—a Langevin-type diffusion driven by score-based generative priors—this time formulated directly in infinite dimensions rather than in the usual finite-dimensional setting. In particular, we consider the continuous-time SDE

$$dX_t = S_\theta(X_t, \tau; \mu)dt + C\nabla_X \log \rho(y - AX_t)dt + \sqrt{2C}dW_t, \tag{3}$$

where $\rho$ is the noise density, $C$ acts as a preconditioner, $\nabla_X$ denotes the Fréchet derivative with respect to $X$, $W_t$ is a Wiener process on $H$, and $S_\theta(X_t, \tau; \mu)$ is a neural network approximation of the score function

$$S(X, \tau; \mu) = -(1 - e^{-\tau})^{-1}(X - e^{-\tau/2}\mathbb{E}[X_0 | X_\tau = X]), \tag{4}$$

which corresponds to the drift term in the time-reversed SDE of the Hilbert-space-valued forward diffusion

$$dX_\tau = -\frac{1}{2}X_\tau d\tau + \sqrt{C}dW_\tau, \qquad X_0 \sim \mu. \tag{5}$$

There are two important aspects to note here. First, both the Langevin SDE (3) and the forward diffusion (5) are driven by a $C$-Wiener process, where $C$ is trace-class, which is crucial for ensuring that the samples are supported on the Hilbert space. Most of the technical difficulties in infinite dimensions arise from this. Second, although the score is often expressed as $\nabla \log p_\tau$ in finite-dimensional settings, the density $p_\tau$ is not defined in infinite dimensions, since a Lebesgue reference measure does not exist. For this reason, in the following we adopt the conditional expectation representation of the score—or, more precisely, an equivalent formulation derived from it, as stated in the next proposition, which was first proved in [49].

**Proposition 2.1.** *The score* (4) *can be written as*

$$S(X, \tau; \mu) = -e^{\tau/2}\mathbb{E}[C(C_\mu C_\tau^{-1})^{-1}\nabla\Phi(X_0) \mid X_\tau = X] - CC_\tau^{-1}X, \tag{6}$$

*where* $C_\tau = e^{-\tau}C_\mu + (1 - e^{-\tau})C$.

The idea behind samplers like (3) is simple yet powerful. By training $S_\theta(X, \tau; \mu)$ to approximate $S(X, \tau; \mu)$, one can effectively learn potentially complex priors $\mu$—since, once $S_\theta(X, \tau; \mu)$ is known, one can sample from $\mu$ by simulating the backward-in-time dynamics—and then incorporate such priors within a Langevin sampling scheme. What remains less understood, however, is how this

approach extends to the infinite-dimensional setting, particularly in relation to the error introduced by approximating the score and whether the sampler remains stable. In the sections that follow, we address this gap—we prove convergence of (3) to the correct posterior and derive error bounds, along with conditions ensuring a globally bounded convergence error. We also elucidate the role of the preconditioner $C$. Our analysis is divided into two parts: one addressing the case of Gaussian priors, and the other the non-Gaussian case.

## 3    Error Analysis in the Gaussian Setting

We begin our analysis of the continous-time Langevin SDE (3) in the infinite-dimensional setting by examining the case where the prior of $X_0$ is a Gaussian measure. While this case may seem to defeat the purpose of using a score-based generative model to learn a simple prior, it provides illuminating insights, as it allows us to detail the impact of the score approximation error on the stationary distribution of (3), offers a clear interpretation of the infinite-dimensional difficulties, and paves the way for the derivation of an explicit form of the optimal preconditioner (Section 4).

We assume in this section that $\Phi = 0$. The posterior (2) is Gaussian:

$$\pi_y = \mathcal{N}\left(\left[C_\mu^{-1} + \sigma^{-2}A^\top A\right]^{-1}\sigma^{-2}A^\top y, \left[C_\mu^{-1} + \sigma^{-2}A^\top A\right]^{-1}\right).$$

We also assume that both $C$ and $C_\mu$ are diagonal in the basis $(v_j)$, with eigenvalues $(\lambda_j)$ and $(\mu_j)$, respectively. We can make a few remarks:

- In the $(v_j)$ basis, the posterior decomposes into a Gaussian $\pi_y^N$ on the span of the first $N$ observed modes and a product of marginal Gaussian over the unobserved modes $j > N$.

- For the observed modes—i.e., those $j \leq N$ influenced by the data through the forward operator $A$—the distribution is

$$\pi_y^N = \mathcal{N}\left(\left[C_{\mu,N}^{-1} + \sigma^{-2}A_N^\top A_N\right]^{-1}\sigma^{-2}A_N^\top y, \left[C_{\mu,N}^{-1} + \sigma^{-2}A_N^\top A_N\right]^{-1}\right), \text{ with } C_{\mu,N} = \operatorname*{Diag}_{1 \leq j \leq N}\left(\mu_j\right).$$

- For the unobserved modes $j > N$, which lie in the nullspace of $A$, the posterior coincides with the prior: $\pi_y^{(j)} = \mathcal{N}\left(0, \mu_j\right)$.

- The score function is $S(X, \tau; \mu) = -\sum_j s_j(\tau; \mu)X^{(j)}v_j$, with $s_j(\tau; \mu) = \frac{\lambda_j}{e^{-\tau}\mu_j + (1 - e^{-\tau})\lambda_j}$.

The block diagonalization of the system by $(v_j)$ justifies the following assumption on the form of the score approximation error.

**Assumption 1.** *We consider an approximate score $S_\theta(X, \tau; \mu)$ such that*

$$\left\langle S(X, \tau; \mu) - S_\theta(X, \tau; \mu), v_j\right\rangle = \varepsilon_j^a(\tau)X^{(j)} + \varepsilon_j^b(\tau).$$

Define $X^N = \sum_{j=1}^N X^{(j)}v_j$ and similarly let $W_t^N$ denote the projection of $W_t$ onto the first $N$ modes. By Assumption 1, for the observed modes $j \leq N$, the preconditioned Langevin dynamics (3) become

$$dX_t^N = -\left[\operatorname*{Diag}_{1 \leq j \leq N}\left(s_j(\tau; \mu)\right) + \operatorname*{Diag}_{1 \leq j \leq N}\left(\varepsilon_j^a(\tau)\right) + C_N\sigma^{-2}A_N^\top A_N\right]X_t^N dt$$

$$+ \left[C_N\sigma^{-2}A_N^\top y - \operatorname*{Diag}_{1 \leq j \leq N}\left(\varepsilon_j^b(\tau)\right)\right]dt + \sqrt{2C_N}dW_t^N,$$

with $C_N = \operatorname*{Diag}_{1 \leq j \leq N}\left(\lambda_j\right)$. For the unobserved modes $j > N$, we have

$$dX_t^{(j)} = -\left[s_j(\tau; \mu) + \varepsilon_j^a(\tau)\right]X_t^{(j)}dt - \varepsilon_j^b(\tau)dt + \sqrt{2\lambda_j}dW_t^{(j)}.$$

We are now ready to derive the stationary distribution of the continuous-time SDE (3). The following proposition makes explicit the dependence on the score approximation error; its proof is included in Appendix A.1.

**Proposition 3.1.** *The stationary distribution $\check{\pi}_y$ of the preconditioned Langevin diffusion with approximate score in the drift term is the Gaussian measure with mean $\check{m}(\tau) = (\check{m}^N(\tau), (\check{m}_j(\tau))_{j \geq N+1})$ and covariance $\check{v}(\tau) = \check{v}^N(\tau) \oplus \underset{j \geq N+1}{\mathrm{Diag}} (\check{v}_j(\tau))$. For the observed modes $j \leq N$, we have*

$$\check{v}^N(\tau) = \left[ C_N^{-1} \underset{1 \leq j \leq N}{\mathrm{Diag}} (s_j(\tau; \mu)) + \sigma^{-2} A_N^\top A_N + C_N^{-1} \underset{1 \leq j \leq N}{\mathrm{Diag}} (\varepsilon_j^a(\tau)) \right]^{-1}, \tag{7}$$

$$\check{m}^N(\tau) = \check{v}^N(\tau) \left[ \sigma^{-2} A_N^\top y - C_N^{-1} \underset{1 \leq j \leq N}{\mathrm{Diag}} (\varepsilon_j^b(\tau)) \right], \tag{8}$$

*while for the unobserved modes $j > N$, we have*

$$\check{v}_j(\tau) = \left[ \lambda_j^{-1} s_j(\tau; \mu) + \lambda_j^{-1} \varepsilon_j^a(\tau) \right]^{-1}, \qquad \check{m}_j(\tau) = -\check{v}_j(\tau) \lambda_j^{-1} \varepsilon_j^b(\tau). \tag{9}$$

Based on Proposition 3.1, we make a few comments:

- If we have access to the perfect score, that is, $\varepsilon_j^a = \varepsilon_j^b = 0$ for all $j$, then

$$\check{m}(\tau) = \left[ C_\tau^{-1} + \sigma^{-2} A^\top A \right]^{-1} \sigma^{-2} A^\top y \overset{\tau \to 0}{\to} \left[ C_\mu^{-1} + \sigma^{-2} A^\top A \right]^{-1} \sigma^{-2} A^\top y,$$

$$\check{v}(\tau) = \left[ C_\tau^{-1} + \sigma^{-2} A^\top A \right]^{-1} \overset{\tau \to 0}{\to} \left[ C_\mu^{-1} + \sigma^{-2} A^\top A \right]^{-1}.$$

  That is, we recover the posterior $\pi_y$ given the data. It does not depend on the preconditioner $C$.

- The error $\varepsilon_j^a$ can have an impact on the stationary distribution of $X_t^{(j)}$, but as long as it is smaller than $\lambda_j / \mu_j$ (i.e., the relative error in the approximation of the score is small), the impact is small.

- The error $\varepsilon_j^b$ can induce a bias. The bias can be large because the mean of the $j$-th mode marginal of $\check{\pi}_y^{(j)}$ is amplified by $\lambda_j^{-1}$. The preconditioner cannot prevent from this bias.

We can make our analysis more quantitative by presenting mode-by-mode and global convergence error estimates for the preconditioned Langevin sampler in the Gaussian setting. To simplify the discussion, the following theorem is stated by assuming that $A_N^\top A_N$ is diagonal.

**Theorem 3.1.** *We define $p_j = \lambda_j / \mu_j$ for all $j$. Let $\check{\pi}_y^{(j)}$ and $\pi_y^{(j)}$ denote the $j$-th mode marginals of the approximate and true posterior distributions, $\check{\pi}_y$ and $\pi_y$, respectively. Suppose that $p_j^{-1} \varepsilon_j^a(\tau) = O(\tau)$, $\lambda_j^{-1} \varepsilon_j^b(\tau) = O(1)$. Then, for $j \leq N$, the Kullback-Leibler divergence satisfies*

$$\mathrm{D_{KL}} \left( \check{\pi}_y^{(j)} \, \big\| \, \pi_y^{(j)} \right) = \frac{1}{2} \lambda_j^{-2} \varepsilon_j^b(\tau)^2$$

$$- \frac{\lambda_j^{-1} \varepsilon_j^b(\tau)}{1 + \sigma^{-2} \mu_j (A_N^\top A_N)_{jj}} \left( \sigma^{-2} (A_N^\top y)_j - \lambda_j^{-1} \varepsilon_j^b(\tau) \right) \left( (p_j - 1)\tau - p_j^{-1} \varepsilon_j^a(\tau) \right) + O(\tau^2). \tag{10}$$

*For $j > N$, we have $\mathrm{D_{KL}} \left( \check{\pi}_y^{(j)} \, \big\| \, \pi_y^{(j)} \right) = \lambda_j^{-2} \varepsilon_j^b(\tau)^2 \left( \frac{1}{2} + (p_j - 1)\tau - p_j^{-1} \varepsilon_j^a(\tau) \right) + O(\tau^2)$.*

*Proof.* The proof relies on Proposition 3.1 and the fact that the $j$-th mode marginals $\check{\pi}_y^{(j)}$ and $\pi_y^{(j)}$ are Gaussian, $\mathcal{N}(\check{m}_j(\tau), \check{v}_j(\tau))$ and $\mathcal{N}(m_j, v_j)$, respectively, hence the Kullback-Leibler divergence has an explicit form and standard perturbation arguments lead to the desired estimates. Full details are provided in Appendix A.2. $\square$

**Remark 3.1.** *Note that Theorem 3.1 can provide a set of sufficient conditions that ensure that the global convergence error of the sampler is bounded in infinite dimensions: $\sum_j |\lambda_j^{-1} \varepsilon_j^b(\tau)| < \infty$, $|p_j^{-1} \varepsilon_j^a(\tau)| < C_1$, and $|(A_N^\top y)_j| < C_2$, where $C_1, C_2$ do not depend on $j$.*

## 4 The Essence of Preconditioning

We now elucidate the role of the preconditioner $C$ in the infinite-dimensional Gaussian setting introduced in the previous section. We begin with two preliminary remarks:

- In our analysis, $C$ first appears in the forward diffusion (5), whose time-reversal learns the prior, and must be carried through the Langevin sampler (3) to target the correct posterior.

- $C$ cannot be the identity: it must be trace-class to keep the diffusion well-posed and to stabilize the Langevin updates across all modes. Indeed, if $C = \mathrm{Diag}(\lambda_j)$, the drift in the $j$-th mode contains the factor $\lambda_j[e^{-\tau}\mu_j + (1 - e^{-\tau})\lambda_j]^{-1}$, which, unless $\lambda_j$ decays sufficiently fast, blows up like $\mu_j^{-1}$ as $j \to \infty$, making the sampler unstable at fine scales. This is a consequence of the infinite-dimensional setting, where $C_\mu$ must be trace-class.

Since the preconditioner plays a role in the rate of convergence across all posterior modes, it is natural to ask whether there exists a $C$ that ensures a uniform convergence rate for the Langevin sampler. To this aim, in the next propositionwe derive the mean reversion rate $\kappa$ of the preconditioned Langevin dynamics (3); the proof is given in Appendix B.1.

**Proposition 4.1.** *Assume that $A_N^\top A_N$ is diagonal. For the observed modes $j \leq N$, the mean reversion rate is*

$$\kappa^{(j)} = \lambda_j \left( \left[e^{-\tau}\mu_j + (1 - e^{-\tau})\lambda_j\right]^{-1} + \sigma^{-2}(A_N^\top A_N)_{jj} + \lambda_j^{-1}\varepsilon_j^a(\tau) \right), \qquad (11)$$

*while for the unobserved modes $j > N$, $\kappa^{(j)} = \lambda_j \left[ \left[e^{-\tau}\mu_j + (1 - e^{-\tau})\lambda_j\right]^{-1} + \lambda_j^{-1}\varepsilon_j^a(\tau)\right].$*

We can make a few comments:
- For the unobserved modes $j > N$, the convergence rate is $\lambda_j[e^{-\tau}\mu_j + (1 - e^{-\tau})\lambda_j]^{-1}$ ($\simeq \lambda_j/\mu_j$ for small $\tau$) when the error $\varepsilon_j^a$ is negligible, and therefore we should choose $\lambda_j = \mu_j$ for all $j$ to get a convergence uniform in $j$, that is to say, $C = C_\mu$.

- For the observed modes $j \leq N$, the convergence rates for those modes such that $\mu_j \ll \sigma^2/(A_N^\top A_N)_{jj}$ (or $(A_N^\top A_N)_{jj} = 0$) are $\lambda_j/\mu_j$, whereas for modes such that $\mu_j \gg \sigma^2/(A_N^\top A_N)_{jj}$ the convergence rates are $\lambda_j\sigma^{-2}(A_N^\top A_N)_{jj}$. We should then choose $\lambda_j = [\mu_j^{-1} + \sigma^{-2}(A_N^\top A_N)_{jj}]^{-1}$, or equivalently $C = [C_\mu^{-1} + \sigma^{-2}A^\top A]^{-1}$.

We now refine our analysis of the preconditioner by incorporating a first-order correction that accounts for the score approximation error at small $\tau$, the regime in which the score is typically learned. The proof of the following theorem relies on a straightforward perturbation argument; full details are given in Appendix B.2.

**Theorem 4.1.** *In addition to Assumption 1, we further suppose that $A_N^\top A_N$ is diagonal, and that $\varepsilon_j^a(\tau) = \varepsilon_j^a\tau + O(\tau^2)$. Under these conditions, the optimal preconditioner $C$ is also diagonal in the basis $(v_j)$, with eigenvalues that admit the expansion $\lambda_j = \lambda_j^{(0)} + \lambda_j^{(1)}\tau + O(\tau^2)$. For the observed modes $j \leq N$, we have*

$$\lambda_j^{(0)} = \left[\mu_j^{-1} + \sigma^{-2}(A_N^\top A_N)_{jj}\right]^{-1}, \qquad \lambda_j^{(1)} = \lambda_j^{(0)\,3}\mu_j^{-2} - \lambda_j^{(0)\,2}\mu_j^{-1} - \varepsilon_j^a\lambda_j^{(0)}. \qquad (12)$$

*For the unobserved modes $j > N$, we have $\lambda_j^{(0)} = \mu_j$, $\lambda_j^{(1)} = -\mu_j\varepsilon_j^a$.*

Based on Theorem 4.1, we make a few comments:

- To compute the preconditioner $C$, one would need information on $A^\top A$, $\sigma$, $C_\mu$, and the score approximation error. Our analysis, however, suggests a simple and practical choice: take $C$ as close as possible to the prior covariance $C_\mu$. For higher-order modes, the leading-order term of the preconditioner coincides with $C_\mu$, and this approximation is particularly justified when the prior decays quickly, so that $\mu_j^{-1} \gg \sigma^{-2}(A_N^\top A_N)$ for the low-order modes. Any available knowledge of the posterior covariance or score error can then be used to refine this first approximation.

- This is not the first occurrence in the literature of an optimal preconditioner for diffusion models in infinite dimensions. For example, while analyzing the convergence error of time-reversed SDE dynamics in infinite dimensions, Pidstrigach et al. [49] derived a similar result for the optimal $C$ by minimizing the Wasserstein-2 distance between the true data distribution and the learned sample distribution. Interestingly, assuming no data model and a perfect score, our framework yields the same optimal $C$, with a few caveats: in our case, the preconditioner arises directly from the mean-reversion rate of the Langevin dynamics. Hence, the optimal covariance we identify does not merely minimize an upper bound: it represents, under the stated assumptions, the best achievable choice in practice for ensuring a uniform rate of convergence across all modes.

## 5 Non-Gaussian Sampling

We can generalize the results of Sections 3 and 4 by considering the case of a general class of prior measures $\mu$ assumed to be absolutely continuous with respect to a Gaussian reference measure $\mathcal{N}(0, C_\mu)$ with density proportional to $\exp(-\Phi)$. We present the main ideas here, relegating the more quantitative results and proofs in Appendix C.

To reproduce the approach of the Gaussian setting, one first needs to diagonalize the Langevin SDE system, which in turn requires diagonalizing the score function $S(X, \tau; \mu)$.

**Proposition 5.1.** *We assume that $C$ and $C_\mu$ have the same basis of eigenfunctions $(v_j)$ and we define $X^{(j)} = \langle X, v_j \rangle$. We assume that $\Phi(X) = \sum_j \phi_j(X^{(j)})$. The score function* (6) *can be written as $S(X, \tau; \mu) = \sum_j S^{(j)}(X^{(j)}, \tau; \mu)v_j$, where*

$$S^{(j)}(X^{(j)}, \tau; \mu) = -\lambda_j \partial_j \check{\phi}_j(X^{(j)}, \tau) - s_j(\tau, \mu)X^{(j)}, \tag{13}$$

*with $\check{\phi}_j(X^{(j)}, \tau) = -\log \mathbb{E}\big[\exp(-\phi_j(\widetilde{X}_0^{(j)})) \mid \widetilde{X}_\tau^{(j)} = X^{(j)}\big]$, and*

$$\begin{pmatrix} \widetilde{X}_0^{(j)} \\ \widetilde{X}_\tau^{(j)} \end{pmatrix} \sim \mathcal{N}\left(0, \begin{pmatrix} \mu_j & e^{-\tau/2}\mu_j \\ e^{-\tau/2}\mu_j & e^{-\tau}\mu_j + (1 - e^{-\tau})\lambda_j \end{pmatrix}\right). \tag{14}$$

We assume a more general form of the score approximation error.

**Assumption 2.** *We consider an approximate score $S_\theta(X, \tau; \mu)$ such that*

$$\big\langle S(X, \tau; \mu) - S_\theta(X, \tau; \mu),\ v_j \big\rangle = \varepsilon_j^a(\tau)\big[X^{(j)} + \partial_j \phi_j(X^{(j)})\big] + \varepsilon_j^b(\tau).$$

With the learned score, the preconditioned Langevin SDE for the observed modes $j \leq N$ becomes

$$dX_t^N = -M_N X_t^N dt + b_N dt + \sqrt{2C_N} dW_t^N,$$

where

$$M_N = \underset{1 \leq j \leq N}{\mathrm{Diag}}\ (s_j(\tau; \mu)) + C_N \sigma^{-2} A_N^\top A_N + \underset{1 \leq j \leq N}{\mathrm{Diag}}\ \big(\varepsilon_j^a(\tau)\big),$$

$$b_N = C_N \sigma^{-2} A_N^\top y - C_N \underset{1 \leq j \leq N}{\mathrm{Diag}}\ \big(\partial_j \check{\phi}_j\big) - \underset{1 \leq j \leq N}{\mathrm{Diag}}\ \big(\varepsilon_j^a(\tau)\partial_j \phi_j(X_t^{(j)})\big) - \underset{1 \leq j \leq N}{\mathrm{Diag}}\ \big(\varepsilon_j^b(\tau)\big).$$

The SDE for the unobserved modes $j > N$ can be obtained by taking $A_N = 0$ above.

The stationary distribution of the preconditioned Langevin SDE is derived in the following proposition, which makes explicit the dependence on the score approximation error.

**Proposition 5.2.** *Let Assumption 2 hold true. Under the hypotheses of the previous proposition, the preconditioned Langevin with approximate score in the drift term has $\check{\pi}_y$ as its stationary distribution. It is absolutely continuous with respect to $\mathcal{N}(\check{m}(\tau), \check{v}(\tau))$, and is given by the density*

$$\frac{d\check{\pi}_y}{d\mathcal{N}(\check{m}(\tau), \check{v}(\tau))}(X, \tau) \propto \exp\left(-\check{\Phi}(X, \tau)\right). \tag{15}$$

*For the observed modes $j \leq N$, the negative log-density is $\check{\Phi}^N(X^N, \tau) = \sum_{j=1}^N \big[\check{\phi}_j(X^{(j)}, \tau) + \lambda_j^{-1}\varepsilon_j^a(\tau)\phi_j(X_t^{(j)})\big]$, the covariance $\check{v}^N(\tau)$ and mean $\check{m}^N(\tau)$ are given by (7-8). For the unobserved modes $j > N$, the negative log-density is $\check{\Phi}^{(j)}(X^{(j)}, \tau) = \check{\phi}_j(X^{(j)}, \tau) + \lambda_j^{-1}\varepsilon_j^a(\tau)\phi_j(X_t^{(j)})$, the covariance $\check{v}^{(j)}(\tau)$ and mean $\check{m}^{(j)}(\tau)$ are given by (9).*

The interested reader can find in Appendix C a quantitative analysis of this general case, including Theorem C.1, which provides an error analysis analogous to Theorem 3.1 for the Gaussian case. Here we make a few qualitative comments:

- If $\varepsilon_j^a = \varepsilon_j^b = 0$ for all $j$, then $\check{\Phi}(X, \tau) \overset{\tau \to 0}{\Rightarrow} \sum_j \phi_j(X^{(j)})$, $\check{v}(\tau) \overset{\tau \to 0}{\Rightarrow} \big[C_\mu^{-1} + \sigma^{-2}A^\top A\big]^{-1}$, $\check{m}(\tau) \overset{\tau \to 0}{\Rightarrow} \big[C_\mu^{-1} + \sigma^{-2}A^\top A\big]^{-1}\sigma^{-2}A^\top y$, that is to say, we get the posterior given the data.

- The preconditioner influences the convergence rate. Consider the case in which $\phi_j$ is convex, i.e., $\partial_j^2 \phi_j \geq C_{\phi_j} > 0$. In this case, for the unobserved modes $j > N$, the convergence rate of the $j$-th mode is $\simeq \lambda_j[\mu_j^{-1} + C_{\phi_j}]$ for small $\tau$, assuming the error $\varepsilon_j^a$ is negligible. To achieve a convergence rate that is uniform in $j$, we should then choose $\lambda_j = [\mu_j^{-1} + C_{\phi_j}]^{-1}$. For the observed modes $j \leq N$, and assuming $A_N^\top A_N$ is diagonal for simplicity, we should instead choose $\lambda_j = [\mu_j^{-1} + \sigma^{-2}(A_N^\top A_N)_{jj} + C_{\phi_j}]^{-1}$.

- If the error $|\varepsilon_j^a| \ll \lambda_j$, its impact on the stationary distribution of $X_t^{(j)}$ is small. Like the Gaussian case, the error $\varepsilon_j^b$ can induce a bias. It can be large since $\lambda_j^{-1} \to +\infty$ as $j \to +\infty$.

## 6 Illustrations

We verify our theory by applying the preconditioned Langevin dynamics with SGM (3) to two linear inverse problems: one based on the Karhunen-Loève (KL) expansion of the Brownian sheet [50], and the other one on an inverse source problem for the heat equation [3]. Both examples are consistent with the theory of the paper. Implementation and further details are provided in Appendix D.

**Brownian sheet.** We illustrate the discretization-invariance property of our approach on the Brownian sheet, represented by its truncated KL expansion $B^N(x) = \sum_{j,k=1}^N \phi_{j,k}(x)\eta_{j,k}$, $x \in [0,1]^2$, where $\eta_{j,k} \sim \mathcal{N}(0,\mu_{j,k})$ and $(\phi_{j,k}, \mu_{j,k})$ are the KL eigenpairs. For $M \leq N$, the inverse problem consists of reconstructing the KL coefficients and Brownian sheet from noisy data $y_{j,k} = \tilde{\eta}_{j,k} + \varepsilon_{j,k}$, $j, k \leq M$, with $\tilde{\eta}_{j,k} \sim \mathcal{N}(0,\mu_{j,k})$, $\varepsilon_{j,k} \sim \mathcal{N}(0,0.01^2)$ (i.e. $A_{jk} = 1$ if $j = k \leq M$ and 0 otherwise). Figure 2 shows the robustness of our approach with respect to the number of modes $M^2$.

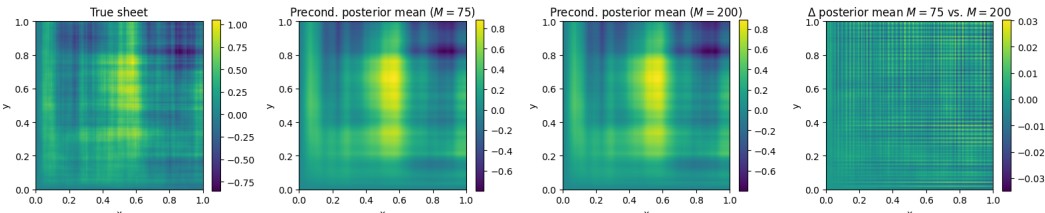

Figure 2: Discretization invariance in reconstructing the KL expansion of the Brownian sheet for increasing number of observed modes $M^2$, with $M = 75$ and $M = 200$. Here $N = 200$.

**Heat equation.** We verify the benefits of the optimal preconditioner $C$ from Theorem 4.1 by considering the ill-posed inverse problem of recovering the initial condition $u(x,0)$, $x \in [0,1]^2$, of the heat equation from noisy observations of the solution $u(x,T)$ at time $T = 0.1$. Expanding in the eigenpairs $(\psi_{j,k}, \zeta_{j,k})$ of the Dirichlet Laplacian, one finds that $u(x,t) = \sum_{j,k} e^{-\zeta_{j,k}t}g_{j,k}\psi_{j,k}(x)$, where $g_{j,k} = \langle u(\cdot,0), \psi_{j,k}\rangle$. The inverse problem diagonalizes: we observe $y_{j,k} = e^{-\zeta_{j,k}T}g_{j,k} + \varepsilon_{j,k}$, $j, k \leq M$, with $g_{j,k} \sim \mathcal{N}(0, e^{-\beta\zeta_{j,k}})$, $\beta = 0.1$, $\varepsilon_{j,k} \sim \mathcal{N}(0,0.005^2)$. In Figure 3 we compare reconstructions using Langevin dynamics preconditioned with the optimal $C$ (3rd column) and vanilla Langevin (4th column). Both samplers use a score perturbed by a relative error $\varepsilon_j^a \sim \mathcal{N}(0,0.1^2)$ scaled by a small $\tau$, as assumed in Theorem 4.1. The results support our theory: (i) the preconditioned sampler is robust to score approximation error, as expected from the design of $C$ (Theorem 4.1); ii) as shown in the autocorrelation plot in Figure 4 (corresponding to the top row of Figure 3), the modes converge faster and more uniformly than with the vanilla dynamics, since $C$ targets the optimal mean reversion rates (Proposition 4.1); iii) vanilla Langevin deteriorates as the score error increases (Figure 3) due to amplification at fine scales, reproducing the pathological behavior seen in Figure 1.

## 7 Discussion and Future Work

We studied a popular sampler—a Langevin-type diffusion driven by score-based generative priors—directly in the infinite-dimensional Bayesian setting, rather than in the usual finite-dimensional one. We showed that naïvely applying standard techniques in infinite dimensions leads to several issues. To ensure provable posterior sampling and discretization-invariance, our analysis shows that

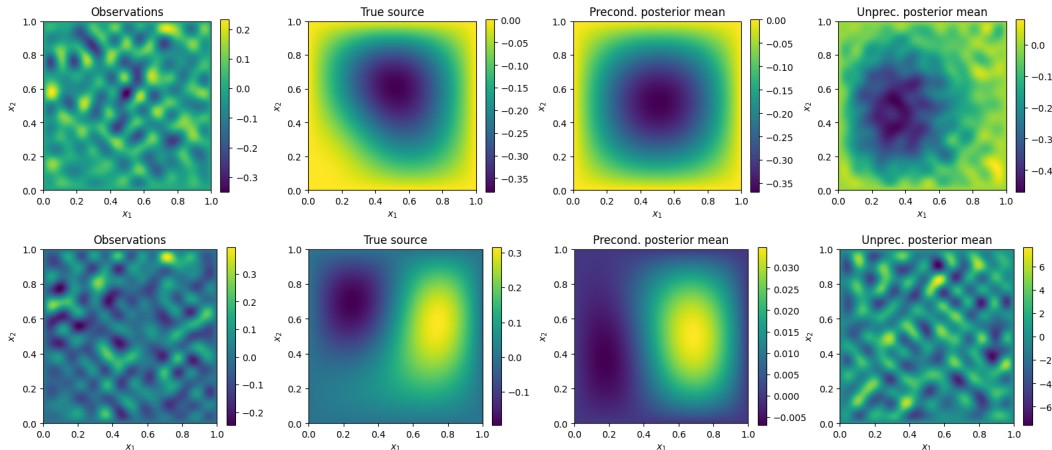

Figure 3: Effects of preconditioning in Langevin sampling for the inverse heat source problem. $M = 15$; top row: $\tau = 10^{-3}$, bottom row: $\tau = 10^{-1}$. The 4th column shows the issues of Figure 1.

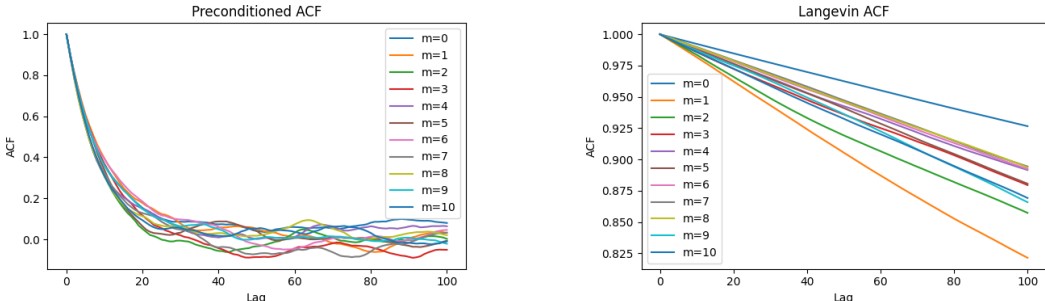

Figure 4: Mode autocorrelation. Left: Preconditioned Langevin with SGM. Right: Vanilla Langevin.

preconditioning the vanilla Langevin is necessary. We prove detailed convergence error estimates and the existence (and form) of an optimal preconditioner—depending on both the forward map $A$ and the score error—that yields uniform convergence across all modes.

As is standard in infinite-dimensional analysis, our results rely on some simplifying assumptions: finite-dimensional data, and co-diagonalizability of the prior and the diffusion's noise covariance. In some parts, we also assumed that $A^\top A$ is diagonal, a common assumption in the theory of linear Bayesian inverse problems [3, 4]. This is not merely a technical convenience: in many classical linear inverse problems, such as the heat equation, tomography, or inverse scattering for Schrödinger-type operators under the Born approximation, the forward operator $A$ is compact, and hence one can always find a basis in which $A^\top A$ is diagonal.

Nevertheless, the main conclusions of our analysis remain valid even without the diagonalization assumption. For example, the asymptotic expansion in Eq. (10) of Theorem 3.1 can be extended to non-diagonal by replacing scalar expansions with their corresponding matrix-series counterparts, while the arguments for the higher-order modes remains the same. Likewise, in Section 4, one can verify that under a perfect score function the optimal preconditioner still takes the form $C = [C_\mu^{-1} + \sigma^{-2} A^\top A]^{-1}$.

Several open questions remain. In particular, how do these results extend to nonlinear inverse problems? Extending the analysis of [21] to determine an optimal preconditioner for nonlinear inverse problems represents an important direction for future work.

## Acknowledgments

JG was supported by Agence de l'Innovation de Défense – AID - via Centre Interdisciplinaire d'Etudes pour la Défense et la Sécurité – CIEDS - (project 2021 - PRODIPO). KS was supported by Air Force Office of Scientific Research under grant FA9550-22-1-0176 and the National Science Foundation under grant DMS-2308389. MVdH acknowledges support from the Department of Energy under grant DE-SC0020345, Oxy, the corporate members of the Geo-Mathematical Imaging Group at Rice University, and the Simons Foundation under the MATH + X program.

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

# A Proofs of Section 3

## A.1 Proof of Proposition 3.1

By Assumption 1, for the observed modes $j \leq N$, the preconditioned Langevin dynamics (3) reduces to the SDE

$$
dX_t^N = - \left[ \operatorname*{Diag}_{1 \leq j \leq N} (s_j(\tau; \mu)) + \operatorname*{Diag}_{1 \leq j \leq N} (\varepsilon_j^a(\tau)) + C_N \sigma^{-2} A_N^\top A_N \right] X_t^N dt
$$
$$
+ \left[ C_N \sigma^{-2} A_N^\top y - \operatorname*{Diag}_{1 \leq j \leq N} (\varepsilon_j^b(\tau)) \right] dt + \sqrt{2C_N} dW_t^N, \tag{16}
$$

with $C_N = \operatorname*{Diag}_{1 \leq j \leq N} (\lambda_j)$. For each unobserved mode $j > N$, we have

$$
dX_t^{(j)} = - \left[ s_j(\tau; \mu) + \varepsilon_j^a(\tau) \right] X_t^{(j)} dt - \varepsilon_j^b(\tau) dt + \sqrt{2\lambda_j} dW_t^{(j)}. \tag{17}
$$

Both (16) and (17) are Ornstein-Uhlenbeck (OU) processes. In particular, $X_t^N \overset{t \to \infty}{\Rightarrow} X_\infty^N$ in distribution, where the distribution of $X_\infty^N$ is the stationary distribution of (16):

$$
X_\infty^N \sim \mathcal{N} \left( \check{m}^N(\tau), \check{v}^N(\tau) \right),
$$

with

$$
\check{m}^N(\tau)
$$
$$
= \left[ \operatorname*{Diag}_{1 \leq j \leq N} (s_j(\tau; \mu)) + \operatorname*{Diag}_{1 \leq j \leq N} (\varepsilon_j^a(\tau)) + C_N \sigma^{-2} A_N^\top A_N \right]^{-1} \left[ C_N \sigma^{-2} A_N^\top y - \operatorname*{Diag}_{1 \leq j \leq N} (\varepsilon_j^b(\tau)) \right]
$$
$$
= \left[ C_N^{-1} \operatorname*{Diag}_{1 \leq j \leq N} (s_j(\tau; \mu)) + C_N^{-1} \operatorname*{Diag}_{1 \leq j \leq N} (\varepsilon_j^a(\tau)) + \sigma^{-2} A_N^\top A_N \right]^{-1} \left[ \sigma^{-2} A_N^\top y - C_N^{-1} \operatorname*{Diag}_{1 \leq j \leq N} (\varepsilon_j^b(\tau)) \right],
$$

and $\check{v}^N(\tau)$ is such that it solves the Lyapunov equation

$$
\left[ \operatorname*{Diag}_{1 \leq j \leq N} (s_j(\tau; \mu)) + \operatorname*{Diag}_{1 \leq j \leq N} (\varepsilon_j^a(\tau)) + C_N \sigma^{-2} A_N^\top A_N \right] \check{v}^N(\tau)
$$
$$
+ \check{v}^N(\tau) \left[ \operatorname*{Diag}_{1 \leq j \leq N} (s_j(\tau; \mu)) + \operatorname*{Diag}_{1 \leq j \leq N} (\varepsilon_j^a(\tau)) + C_N \sigma^{-2} A_N^\top A_N \right]^\top = 2C_N.
$$

Then

$$
\check{v}^N(\tau) = \left[ C_N^{-1} \operatorname*{Diag}_{1 \leq j \leq N} (s_j(\tau; \mu)) + C_N^{-1} \operatorname*{Diag}_{1 \leq j \leq N} (\varepsilon_j^a(\tau)) + \sigma^{-2} A_N^\top A_N \right]^{-1},
$$

since

$$
\left[ \operatorname*{Diag}_{1 \leq j \leq N} (s_j(\tau; \mu)) + \operatorname*{Diag}_{1 \leq j \leq N} (\varepsilon_j^a(\tau)) + C_N \sigma^{-2} A_N^\top A_N \right] \check{v}^N(\tau)
$$
$$
= C_N \left( \check{v}^N(\tau) \right)^{-1} \check{v}^N(\tau) = C_N,
$$

and

$$
\check{v}^N(\tau) \left[ C_N^{-1} \operatorname*{Diag}_{1 \leq j \leq N} (s_j(\tau; \mu)) + C_N^{-1} \operatorname*{Diag}_{1 \leq j \leq N} (\varepsilon_j^a(\tau)) + \sigma^{-2} A_N^\top A_N \right] C_N
$$
$$
= \check{v}^N(\tau) \left( \check{v}^N(\tau) \right)^{-1} C_N = C_N.
$$

For each $j > N$, (17) is a one-dimensional OU process with rate $s_j(\tau; \mu) + \varepsilon_j^a(\tau)$, mean shift $-[s_j(\tau; \mu) + \varepsilon_j^a(\tau)]^{-1} \varepsilon_j^b(\tau)$, and noise $\sqrt{2\lambda_j}$. Hence $X_t^{(j)} \overset{t \to \infty}{\Rightarrow} X_\infty^{(j)}$ in distribution, where the distribution of $X_\infty^{(j)}$ is the stationary distribution of (17)

$$
X_\infty^{(j)} \sim \mathcal{N} \left( -\frac{\varepsilon_j^b(\tau)}{s_j(\tau; \mu) + \varepsilon_j^a(\tau)}, \frac{\lambda_j}{s_j(\tau; \mu) + \varepsilon_j^a(\tau)} \right).
$$

These results are valid as soon as $s_j(\tau; \mu) + \varepsilon_j^a(\tau)$ is a positive number for all $j$.

## A.2 Proof of Theorem 3.1

For each mode $j$, define

$$\check{\mu}_j = \mu_j \left[ e^{-\tau} + (1 - e^{-\tau}) p_j \right].$$

The proof can be divided into two cases: one for the observed modes $j \leq N$, and one for the unobserved modes $j > N$. Since the KL divergence for the unobserved modes can be obtained by taking $A_N = 0$ in the expression for the observed modes, we focus only on the latter.

The marginal distributions of the $j$-th mode for the approximate and true posterior, for $j \leq N$, are given respectively by

$$\check{\pi}_y^{(j)} = \mathcal{N} \left( \left[ \frac{1}{\check{\mu}_j} + \sigma^{-2}(A_N^\top A_N)_{jj} + \lambda_j^{-1} \varepsilon_j^a(\tau) \right]^{-1} \left[ \sigma^{-2}(A_N^\top y)_j - \lambda_j^{-1} \varepsilon_j^b(\tau) \right], \right.$$

$$\left. \left[ \frac{1}{\check{\mu}_j} + \sigma^{-2}(A_N^\top A_N)_{jj} + \lambda_j^{-1} \varepsilon_j^a(\tau) \right]^{-1} \right),$$

and

$$\pi_y^{(j)} = \mathcal{N} \left( \left[ \frac{1}{\mu_j} + \sigma^{-2}(A_N^\top A_N)_{jj} \right]^{-1} \sigma^{-2}(A_N^\top y)_j, \left[ \frac{1}{\mu_j} + \sigma^{-2}(A_N^\top A_N)_{jj} \right]^{-1} \right).$$

The KL divergence between these two Gaussian distributions admits the explicit formula

$$D_{\mathrm{KL}} \left( \check{\pi}_y^{(j)} \,\middle\|\, \pi_y^{(j)} \right)$$

$$= \log \left( \left[ \frac{1}{\mu_j} + \sigma^{-2}(A_N^\top A_N)_{jj} \right]^{-1} \left[ \frac{1}{\check{\mu}_j} + \sigma^{-2}(A_N^\top A_N)_{jj} + \lambda_j^{-1} \varepsilon_j^a(\tau) \right] \right)$$

$$- \frac{1}{2} + \frac{1}{2} \left[ \frac{1}{\mu_j} + \sigma^{-2}(A_N^\top A_N)_{jj} \right]^2 \left\{ \left[ \frac{1}{\check{\mu}_j} + \sigma^{-2}(A_N^\top A_N)_{jj} + \lambda_j^{-1} \varepsilon_j^a(\tau) \right]^{-2} \right.$$

$$+ \left( \left[ \frac{1}{\check{\mu}_j} + \sigma^{-2}(A_N^\top A_N)_{jj} + \lambda_j^{-1} \varepsilon_j^a(\tau) \right]^{-1} \left[ \sigma^{-2}(A_N^\top y)_j - \lambda_j^{-1} \varepsilon_j^b(\tau) \right] \right.$$

$$\left. \left. - \left[ \frac{1}{\mu_j} + \sigma^{-2}(A_N^\top A_N)_{jj} \right]^{-1} \sigma^{-2}(A_N^\top y)_j \right)^2 \right\}.$$

We study its limiting behavior as $\tau \to 0$. From the first term, we derive

$$\log \left( \left[ \frac{1}{\mu_j} + \sigma^{-2}(A_N^\top A_N)_{jj} \right]^{-1} \left[ \frac{1}{\check{\mu}_j} + \sigma^{-2}(A_N^\top A_N)_{jj} + \lambda_j^{-1} \varepsilon_j^a(\tau) \right] \right)$$

$$= \log \left( \left[ 1 + \sigma^{-2} \mu_j (A_N^\top A_N)_{jj} \right]^{-1} \left[ \frac{1}{e^{-\tau} + (1 - e^{-\tau}) p_j} + \sigma^{-2} \mu_j (A_N^\top A_N)_{jj} + p_j^{-1} \varepsilon_j^a(\tau) \right] \right)$$

$$= \log \left( 1 + \frac{1}{1 + \sigma^{-2} \mu_j (A_N^\top A_N)_{jj}} \left[ \frac{(1 - e^{-\tau}) + (e^{-\tau} - 1) p_j}{e^{-\tau} + (1 - e^{-\tau}) p_j} + p_j^{-1} \varepsilon_j^a(\tau) \right] \right)$$

$$= \log \left( 1 + \frac{1}{1 + \sigma^{-2} \mu_j (A_N^\top A_N)_{jj}} \left[ -\tau(p_j - 1) + p_j^{-1} \varepsilon_j^a(\tau) + O(\tau^2) \right] \right)$$

$$= \frac{1}{1 + \sigma^{-2} \mu_j (A_N^\top A_N)_{jj}} \left( -\tau(p_j - 1) + p_j^{-1} \varepsilon_j^a(\tau) \right) + O(\tau^2). \tag{18}$$

We now consider the other terms of the KL divergence. First, we derive

$$
\left[\frac{1}{\breve{\mu}_j} + \sigma^{-2}(A_N^\top A_N)_{jj} + \lambda_j^{-1}\varepsilon_j^a(\tau)\right]^{-1}
$$

$$
\times \left[\sigma^{-2}(A_N^\top y)_j - \lambda_j^{-1}\varepsilon_j^b(\tau)\right] - \left[\frac{1}{\mu_j} + \sigma^{-2}(A_N^\top A_N)_{jj}\right]^{-1}\sigma^{-2}(A_N^\top y)_j
$$

$$
= \frac{\mu_j}{1+\sigma^{-2}\mu_j(A_N^\top A_N)_{jj}}\left\{\left[1 + \frac{1}{1+\sigma^{-2}\mu_j(A_N^\top A_N)_{jj}}\left(\frac{(1-e^{-\tau})+(e^{-\tau}-1)p_j}{e^{-\tau}+(1-e^{-\tau})p_j}\right.\right.\right.
$$

$$
\left.\left.\left. + p_j^{-1}\varepsilon_j^a(\tau)\right)\right]^{-1}\left[\sigma^{-2}(A_N^\top y)_j - \lambda_j^{-1}\varepsilon_j^b(\tau)\right] - \sigma^{-2}(A_N^\top y)_j\right\}
$$

$$
= \frac{\mu_j}{1+\sigma^{-2}\mu_j(A_N^\top A_N)_{jj}}\left\{\left[1 + \frac{1}{1+\sigma^{-2}\mu_j(A_N^\top A_N)_{jj}}\left(-\tau(p_j-1)+p_j^{-1}\varepsilon_j^a(\tau)+O(\tau^2)\right)\right]^{-1}\right.
$$

$$
\left. \times \left[\sigma^{-2}(A_N^\top y)_j - \lambda_j^{-1}\varepsilon_j^b(\tau)\right] - \sigma^{-2}(A_N^\top y)_j\right\}
$$

$$
= \frac{\mu_j}{1+\sigma^{-2}\mu_j(A_N^\top A_N)_{jj}}\left\{\left[1 + \frac{1}{1+\sigma^{-2}\mu_j(A_N^\top A_N)_{jj}}\left(\tau(p_j-1)-p_j^{-1}\varepsilon_j^a(\tau)+O(\tau^2)\right)\right]\right.
$$

$$
\left. \times \left[\sigma^{-2}(A_N^\top y)_j - \lambda_j^{-1}\varepsilon_j^b(\tau)\right] - \sigma^{-2}(A_N^\top y)_j\right\}
$$

$$
= \frac{\mu_j}{1+\sigma^{-2}\mu_j(A_N^\top A_N)_{jj}}\left\{-\lambda_j^{-1}\varepsilon_j^b(\tau) - \frac{\lambda_j^{-1}\varepsilon_j^b(\tau)}{1+\sigma^{-2}\mu_j(A_N^\top A_N)_{jj}}\left(\tau(p_j-1)-p_j^{-1}\varepsilon_j^a(\tau)\right)\right.
$$

$$
\left. + \frac{\sigma^{-2}(A_N^\top y)_j}{1+\sigma^{-2}\mu_j(A_N^\top A_N)_{jj}}\left(\tau(p_j-1)-p_j^{-1}\varepsilon_j^a(\tau)\right)\right\} + O(\tau^2).
$$

Taking the square yields

$$
\left(\left[\frac{1}{\breve{\mu}_j} + \sigma^{-2}(A_N^\top A_N)_{jj} + \lambda_j^{-1}\varepsilon_j^a(\tau)\right]^{-1}\right.
$$

$$
\left. \times \left[\sigma^{-2}(A_N^\top y)_j + \lambda_j^{-1}\varepsilon_j^b(\tau)\right] - \left[\frac{1}{\mu_j} + \sigma^{-2}(A_N^\top A_N)_{jj}\right]^{-1}\sigma^{-2}(A_N^\top y)_j\right)^2
$$

$$
= \frac{\mu_j^2}{(1+\sigma^{-2}\mu_j(A_N^\top A_N)_{jj})^2}\left[\lambda_j^{-2}\varepsilon_j^b(\tau)^2\right.
$$

$$
\left. + \frac{2\lambda_j^{-1}\varepsilon_j^b(\tau)}{1+\sigma^{-2}\mu_j(A_N^\top A_N)_{jj}}\left(\lambda_j^{-1}\varepsilon_j^b(\tau)-\sigma^{-2}(A_N^\top y)_j\right)\left(\tau(p_j-1)-p_j^{-1}\varepsilon_j^a(\tau)\right)\right] + O(\tau^2).
$$

$$(19)$$

Next, we consider

$$
\left[\frac{1}{\breve{\mu}_j} + \sigma^{-2}(A_N^\top A_N)_{jj} + \lambda_j^{-1}\varepsilon_j^a(\tau)\right]^{-2}
$$

$$
= \frac{\mu_j^2}{(1+\sigma^{-2}\mu_j(A_N^\top A_N)_{jj})^2}\left[1 + \frac{1}{1+\sigma^{-2}\mu_j(A_N^\top A_N)_{jj}}\left(-\tau(p_j-1)+p_j^{-1}\varepsilon_j^a(\tau)+O(\tau^2)\right)\right]^{-2}
$$

$$
= \frac{\mu_j^2}{(1+\sigma^{-2}\mu_j(A_N^\top A_N)_{jj})^2}\left[1 + \frac{2}{1+\sigma^{-2}\mu_j(A_N^\top A_N)_{jj}}\left(\tau(p_j-1)-p_j^{-1}\varepsilon_j^a(\tau)\right)+O(\tau^2)\right].
$$

$$(20)$$

Putting (18), (19), and (20) together, we obtain

$$D_{KL}\left(\check{\pi}_y^{(j)} \,\|\, \pi_y^{(j)}\right) = \frac{1}{2}\lambda_j^{-2}\varepsilon_j^b(\tau)^2$$

$$- \frac{\lambda_j^{-1}\varepsilon_j^b(\tau)}{1 + \sigma^{-2}\mu_j(A_N^\top A_N)_{jj}}\left(\sigma^{-2}(A_N^\top y)_j - \lambda_j^{-1}\varepsilon_j^b(\tau)\right)\left(\tau(p_j - 1) - p_j^{-1}\varepsilon_j^a(\tau)\right) + O(\tau^2).$$

# B  Proofs of Section 4

## B.1  Proof of Proposition 4.1

If we assume that $A_N^\top A_N$ is diagonal in $(v_j)$, for the observed modes $j \leq N$ the preconditioned Langevin dynamics 3 becomes

$$dX_t^{(j)} = -\left[s_j(\tau;\mu) + \varepsilon_j^a(\tau) + \lambda_j\sigma^{-2}(A_N^\top A_N)_{jj}\right]X_t^{(j)}dt - \left[\lambda_j\sigma^{-2}(A_N^\top y)_j - \varepsilon_j^b(\tau)\right]dt$$
$$+ \sqrt{2\lambda_j}dW_t^{(j)}, \tag{21}$$

while for each unobserved mode $j > N$, one obtains

$$dX_t^{(j)} = -\left[s_j(\tau;\mu) + \varepsilon_j^a(\tau)\right]X_t^{(j)}dt - \varepsilon_j^b(\tau)dt + \sqrt{2\lambda_j}dW_t^{(j)}. \tag{22}$$

Let $m^{(j)}(t) = \mathbb{E}[X_t^{(j)}]$. For the observed modes $j \leq N$, taking the expectation in the SDE above gives the linear ODEs

$$\frac{dm^{(j)}}{dt} = -\left[s_j(\tau;\mu) + \varepsilon_j^a(\tau) + \lambda_j\sigma^{-2}(A_N^\top A_N)_{jj}\right]m^{(j)}(t) + \left[\lambda_j\sigma^{-2}(A_N^\top y)_j - \varepsilon_j^b(\tau)\right]$$

while for each unobserved mode $j > N$ one obtains

$$\frac{dm^{(j)}}{dt} = -\left[s_j(\tau;\mu) + \varepsilon_j^a(\tau)\right]m^{(j)}(t) - \varepsilon_j^b(\tau).$$

Given $m^{(j)}(0) = m_0^{(j)}$, both have unique solution. For $j \leq N$,

$$m^{(j)}(t) = \left(m^{(j)}(0) - \frac{\lambda_j\sigma^{-2}(A_N^\top y)_j - \varepsilon_j^b(\tau)}{s_j(\tau;\mu) + \varepsilon_j^a(\tau) + \lambda_j\sigma^{-2}(A_N^\top A_N)_{jj}}\right)e^{-\left[s_j(\tau;\mu)+\varepsilon_j^a(\tau)+\lambda_j\sigma^{-2}(A_N^\top A_N)_{jj}\right]t}$$

$$+ \frac{\lambda_j\sigma^{-2}(A_N^\top y)_j - \varepsilon_j^b(\tau)}{s_j(\tau;\mu) + \varepsilon_j^a(\tau) + \lambda_j\sigma^{-2}(A_N^\top A_N)_{jj}},$$

which for $t \to \infty$ decays exponentially fast to the mean

$$\frac{\lambda_j\sigma^{-2}(A_N^\top y)_j - \varepsilon_j^b(\tau)}{s_j(\tau;\mu) + \varepsilon_j^a(\tau) + \lambda_j\sigma^{-2}(A_N^\top A_N)_{jj}},$$

with rate

$$\kappa^{(j)} = s_j(\tau;\mu) + \varepsilon_j^a(\tau) + \lambda_j\sigma^{-2}(A_N^\top A_N)_{jj}$$
$$= \lambda_j\left(\left[e^{-\tau}\mu_j + (1 - e^{-\tau})\lambda_j\right]^{-1} + \sigma^{-2}(A_N^\top A_N)_{jj} + \lambda_j^{-1}\varepsilon_j^a(\tau)\right).$$

For the unobserved modes $j > N$,

$$m^{(j)}(t) = \left(m^{(j)}(0) + \frac{\varepsilon_j^b(\tau)}{s_j(\tau;\mu) + \varepsilon_j^a(\tau)}\right)e^{-\left[s_j(\tau;\mu)+\varepsilon_j^a(\tau)\right]t} - \frac{\varepsilon_j^b(\tau)}{s_j(\tau;\mu) + \varepsilon_j^a(\tau)},$$

which converge to the mean

$$-\frac{\varepsilon_j^b(\tau)}{s_j(\tau;\mu) + \varepsilon_j^a(\tau)},$$

with rate

$$\kappa^{(j)} = s_j(\tau;\mu) + \varepsilon_j^a(\tau)$$
$$= \lambda_j\left[\left[e^{-\tau}\mu_j + (1 - e^{-\tau})\lambda_j\right]^{-1} + \lambda_j^{-1}\varepsilon_j^a(\tau)\right].$$

## B.2 Proof of Theorem 4.1

We consider only the case of the observed modes $j \leq N$, since the case for the unobserved modes $j > N$ follows directly by setting $A_N = 0$.

By Proposition 4.1, ensuring uniform convergence rate for (3) using an approximate score function—as described in Assumption 1—amounts to solving the equation

$$\frac{\lambda_j}{\check{\mu}_j} + \lambda_j \sigma^{-2}(A_N^\top A_N)_{jj} + \varepsilon_j^a(\tau) = 1, \tag{23}$$

where $\check{\mu}_j = \mu_j \left[ e^{-\tau} + (1 - e^{-\tau})p_j \right]$.

Assume the expansions

$$\lambda_j = \lambda_j^{(0)} + \lambda_j^{(1)}\tau + O(\tau^2), \qquad \varepsilon_j^a(\tau) = \varepsilon_j^a \tau + O(\tau^2).$$

Then we compute

$$\check{\mu}_j = \mu_j + \mu_j(p_j - 1)\tau + O(\tau^2) = \mu_j + \lambda_j^{(0)}\tau - \mu_j\tau + O(\tau^2),$$

using that $p_j = \lambda_j / \mu_j$ Substituting into (23), we obtain that $\lambda_j^{(0)}$ and $\lambda_j^{(1)}$ must satisfy

$$\mu_j^{-1}(\lambda_j^{(0)} + \lambda_j^{(0)}[1 - \mu_j^{-1}\lambda_j^{(0)}]\tau + \lambda_j^{(1)}\tau) + \sigma^{-2}(A_N^\top A_N)_{jj}(\lambda_j^{(0)} + \lambda_j^{(1)}\tau) + \varepsilon_j^a\tau + O(\tau^2) = 1.$$

Rearranging the terms, we get $\lambda_j^{(0)}\mu_j^{-1} + \lambda_j^{(0)}\sigma^{-2}(A_N^\top A_N)_{jj} = 1$, which gives

$$\lambda_j^{(0)} = \left[ \mu_j^{-1} + \sigma^{-2}(A_N^\top A_N)_{jj} \right]^{-1},$$

and

$$\mu_j^{-1}\left( -\lambda_j^{(0)}\left[ \mu_j^{-1}\lambda_j^{(0)} - 1 \right] + \lambda_j^{(1)} \right) + \sigma^{-2}(A_N^\top A_N)_{jj}\lambda_j^{(1)} + \varepsilon_j^a = 0,$$

yielding

$$\lambda_j^{(1)} = \lambda_j^{(0)} \left( \mu_j^{-1}\lambda_j^{(0)} \left( \mu_j^{-1}\lambda_j^{(0)} - 1 \right) - \varepsilon_j^a \right).$$

## C  Non-Gaussian Sampling: Technical Details

### C.1  Proof of Proposition 5.1

By $\Phi(X) = \sum_j \phi_j(X^{(j)})$, the prior $\mu$ has Radon-Nikodym derivative with respect to the Gaussian $\mathcal{N}(0, C_\mu)$ given by

$$\frac{d\mu}{d\mathcal{N}(0, C_\mu)}(X) \propto \prod_j \exp\left( -\phi_j(X^{(j)}) \right).$$

Since $C$ and $C_\mu$ are both diagonalized by the same basis $(v_j)$, the prior factorizes as a product of independent one-dimensional marginals in the coordinates $X^{(j)}$:

$$\frac{d\mu}{d\mathcal{N}(0, C_\mu)}(X) = \prod_j \frac{d\mu^{(j)}}{d\mathcal{N}(0, \mu_j)}(X^{(j)}),$$

where

$$\frac{d\mu^{(j)}}{d\mathcal{N}(0, \mu_j)}(X^{(j)}) \propto \exp\left( -\phi_j(X^{(j)}) \right).$$

We can then work mode by mode. For each $j$, define the one-dimensional OU process

$$\widetilde{X}_0^{(j)} \sim \mathcal{N}(0, \mu_j), \qquad \widetilde{X}_\tau^{(j)} = e^{-\tau/2}\widetilde{X}_0^{(j)} + \sqrt{1 - e^{-\tau}}\xi^{(j)},$$

with $\xi^{(j)} \sim \mathcal{N}(0, \lambda_j)$ independent of $\widetilde{X}_0^{(j)}$. Notice that

$$\begin{pmatrix} \widetilde{X}_0^{(j)} \\ \widetilde{X}_\tau^{(j)} \end{pmatrix} \sim \mathcal{N}\left( 0, \begin{pmatrix} \mu_j & e^{-\tau/2}\mu_j \\ e^{-\tau/2}\mu_j & e^{-\tau}\mu_j + (1 - e^{-\tau})\lambda_j \end{pmatrix} \right). \tag{24}$$

The OU transition kernel is

$$\tilde{p}(\widetilde{X}_\tau^{(j)} = x_\tau \mid \widetilde{X}_0^{(j)} = x_0) = \mathcal{N}(e^{-\tau/2}x_0, (1 - e^{-\tau})\lambda_j)(x_\tau),$$

where $\mathcal{N}(\mu, \sigma^2)(x)$ is the density at $x$ of the normal distribution with mean $\mu$ and variance $\sigma^2$. We push forward the prior $e^{-\phi_j}\mathcal{N}(0, \mu_j)$ through the OU kernel. Mode by mode, its density is

$$\mu_\tau^{(j)}(x_\tau) \propto \int \exp(-\phi_j(x_0))\mathcal{N}(0, \mu_j)(x_0)\tilde{p}(x_\tau \mid x_0)dx_0$$

$$= \int \exp(-\phi_j(x_0))\tilde{p}_{0,\tau}(x_0, x_\tau)dx_0,$$

where $\tilde{p}_{0,\tau}(x_0, x_\tau)$ denotes the joint density of $(\widetilde{X}_0^{(j)}, \widetilde{X}_\tau^{(j)})$. Let $\check{\mu}_j = e^{-\tau}\mu_j + (1 - e^{-\tau})\lambda_j$. Dividing by the marginal Gaussian density of $\widetilde{X}_\tau^{(j)}$, we get

$$\frac{\mu_\tau^{(j)}(x_\tau)}{\mathcal{N}(0, \check{\mu}_j)(x_\tau)} \propto \int \exp(-\phi_j(x_0))\frac{\tilde{p}_{0,\tau}(x_0, x_\tau)}{\tilde{p}_\tau(x_\tau)}dx_0$$

$$= \int \exp(-\phi_j(x_0))\tilde{p}(x_0 \mid x_\tau)dx_0$$

$$= \mathbb{E}[\exp(-\phi_j(\widetilde{X}_0^{(j)})) \mid \widetilde{X}_\tau^{(j)} = x_\tau].$$

Since $S^{(j)}(x_\tau, \tau; \mu_j) = \lambda_j\partial_j \log \frac{\mu_\tau^{(j)}(x_\tau)}{\mathcal{N}(0, \check{\mu}_j)} + \lambda_j\mathcal{N}(0, \check{\mu}_j)(x_\tau)$, we obtain

$$S^{(j)}(x_\tau, \tau; \mu) = \lambda_j\partial_j \log \mathbb{E}[\exp(-\phi_j(\widetilde{X}_0^{(j)})) \mid \widetilde{X}_\tau^{(j)} = x_\tau] - \frac{\lambda_j}{e^{-\tau}\mu_j + (1 - e^{-\tau})\lambda_j}X^{(j)}.$$

### C.2 Proof of Proposition 5.2

For each mode $j$, define

$$\check{\phi}_j(X^{(j)}, \tau) = -\log \mathbb{E}\left[\exp(-\phi_j(\widetilde{X}_0^{(j)})) \mid \widetilde{X}_\tau^{(j)} = X^{(j)}\right].$$

Under Assumption 2, the first $N$ coordinates of the preconditioned Langevin dynamics (3) corresponding to the observed modes $j \leq N$ satisfy

$$dX_t^N = -\left[\operatorname*{Diag}_{1\leq j\leq N}(s_j(\tau; \mu)) + C_N\sigma^{-2}A_N^\top A_N + \operatorname*{Diag}_{1\leq j\leq N}(\varepsilon_j^a(\tau))\right]X_t^N dt$$

$$+ \left[C_N\sigma^{-2}A_N^\top y - C_N\operatorname*{Diag}_{1\leq j\leq N}\left(\partial_j\check{\phi}_j(X_t^{(j)})\right) - \operatorname*{Diag}_{1\leq j\leq N}\left(\varepsilon_j^a(\tau)\partial_j\phi_j(X_t^{(j)})\right) \quad (25)$$

$$- \operatorname*{Diag}_{1\leq j\leq N}(\varepsilon_j^b(\tau))\right]dt + \sqrt{2C_N}dW_t^N.$$

The SDE for the unobserved modes $j > N$ can be obtained by taking $A_N = 0$ above:

$$dX_t^{(j)} = -\left[s_j(\tau; \mu) + \varepsilon_j^a(\tau)\right]X_t^{(j)}dt$$
$$+ \left[-\lambda_j\partial_j\check{\phi}_j(X_t^{(j)}) - \varepsilon_j^a(\tau)\partial_j\phi_j(X_t^{(j)}) - \varepsilon_j^b(\tau)\right]dt + \sqrt{2\lambda_j}dW_t^{(j)}. \quad (26)$$

Both (25) and (26) are preconditioned overdamped Langevin SDEs. In particular, one checks that (25) can be written as

$$dX_t^N = -C_N\nabla U_N(X_t^N)dt + \sqrt{2C_N}dW_t^N,$$

where the potential $U_N$ is

$$U_N(X^N) = \frac{1}{2}X^{N\top}\left[C_N^{-1}\operatorname*{Diag}_{1\leq j\leq N}(s_j(\tau; \mu)) + \sigma^{-2}A_N^\top A_N + C_N^{-1}\operatorname*{Diag}_{1\leq j\leq N}(\varepsilon_j^a(\tau))\right]X^N$$

$$- \left[\sigma^{-2}A_N^\top y - C_N^{-1}\operatorname*{Diag}_{1\leq j\leq N}(\varepsilon_j^b(\tau))\right]X^N + \sum_{j=1}^N\left[\check{\phi}_j(X_t^{(j)}) - \lambda_j^{-1}\varepsilon_j^a(\tau)\phi_j(X_t^{(j)})\right].$$

Its stationary distribution is $\check{\pi}_y^N$, which is absolutely continuous with respect to the Lebesgue measure over $\mathbb{R}^N$:

$$\frac{d\check{\pi}_y^N(X^N)}{dX^N} \propto \exp(-U_N(X_N)).$$

We split $U_N$ into quadratic and non-quadratic terms. Hence

$$\frac{d\check{\pi}_y^N(X^N)}{dX^N} \propto \exp\left(-\check{\Phi}^N(X^N, \tau)\right) \mathcal{N}(\check{m}^N(\tau), \check{v}^N(\tau))(X^N),$$

where $\mathcal{N}(\check{m}^N(\tau), \check{v}^N(\tau))(X_N)$ is the density at $X_N$ of the multivariate Gaussian with mean $\check{m}^N(\tau)$ and covariance $\check{v}^N(\tau)$, with

$$\check{\Phi}^N(X^N, \tau) = \sum_{j=1}^N \left[\check{\phi}_j(X^{(j)}, \tau) + \lambda_j^{-1}\varepsilon_j^a\phi_j(X_t^{(j)})\right],$$

$$\check{v}^N(\tau) = \left[C_N^{-1} \operatorname*{Diag}_{1\leq j\leq N}(s_j(\tau; \mu)) + \sigma^{-2}A_N^\top A_N + C_N^{-1} \operatorname*{Diag}_{1\leq j\leq N}(\varepsilon_j^a(\tau))\right]^{-1},$$

$$\check{m}^N(\tau) = \check{v}^N(\tau)\left[\sigma^{-2}A_N^\top y - C_N^{-1} \operatorname*{Diag}_{1\leq j\leq N}(\varepsilon_j^b(\tau))\right].$$

By the same argument, for each $j > N$, the one-dimensional potential of (26) is

$$U_j(X^{(j)}) = \left[\lambda_j^{-1}s_j(\tau; \mu) + \lambda_j^{-1}\varepsilon_j^a(\tau)\right]\frac{X_t^{(j)\,2}}{2} + \lambda_j^{-1}\varepsilon_j^b(\tau)X_t^{(j)} + \check{\phi}_j(X_t^{(j)}) + \lambda_j^{-1}\varepsilon_j^a(\tau)\phi_j(X_t^{(j)}).$$

Its stationary distribution is therefore

$$\frac{d\check{\pi}_y^{(j)}(X^{(j)})}{dX^{(j)}} \propto \exp\left(-\check{\Phi}^{(j)}(X^{(j)}, \tau)\right) \mathcal{N}(\check{m}^{(j)}(\tau), \check{v}^{(j)}(\tau))(X^{(j)}),$$

where $\mathcal{N}(\check{m}^{(j)}(\tau), \check{v}^{(j)}(\tau))(X^{(j)})$ is the density at $X^{(j)}$ of the multivariate Gaussian with mean $\check{m}^{(j)}(\tau)$ and covariance $\check{v}^{(j)}(\tau)$, with

$$\check{\Phi}^{(j)}(X^{(j)}, \tau) = \check{\phi}_j(X^{(j)}, \tau) + \lambda_j^{-1}\varepsilon_j^a\phi_j(X_t^{(j)}),$$

$$\check{v}^{(j)}(\tau) = \left[\lambda_j^{-1}s_j(\tau; \mu) + \lambda_j^{-1}\varepsilon_j^a(\tau)\right]^{-1},$$

$$\check{m}^{(j)}(\tau) = -\check{v}^{(j)}(\tau)\lambda_j^{-1}\varepsilon_j^b(\tau).$$

### C.3 Error Analysis in the Non-Gaussian Setting

For the sake of completeness, we present a result analogous to Theorem 3.1 for the non-Gaussian case. As the interested reader will notice, the calculations are significantly more involved, but remain relatively straightforward.

**Theorem C.1.** *We assume $\varepsilon_j^a(\tau) = O(\tau)$, $\varepsilon_j^b(\tau) = O(1)$, and $A_N = \operatorname{Diag}_{1\leq j\leq N}(A_{jj})$. The Kullback-Leibler divergence between $\check{\pi}_y^{(j)}$ and $\pi_y^{(j)}$ is given by*

$$\mathrm{D_{KL}}\left(\check{\pi}_y^{(j)} \,\middle|\middle|\, \pi_y^{(j)}\right) = B_j(\tau) + E_j(\tau),$$

*where $B_j(\tau)$ is a bias term given by*

$$B_j(\tau) = -2\lambda_j^{-1}\varepsilon_j^b(\tau)\mathbb{E}_{\check{\pi}_y^{(j)}}[x] + \lambda_j^{-1}p_j^{-1}(\varepsilon_j^b(\tau))^2 + \log\int e^{-\phi_j(z) - \frac{1}{2\sigma^2}[A_{jj}z - y_j]^2}\mathcal{N}(0, \mu_j)(z)dz$$

$$- \log\int e^{-\phi_j(z) - \frac{1}{2\sigma^2}[A_{jj}z - y_j]^2}\mathcal{N}(-p_j^{-1}\varepsilon_j^b(\tau), \mu_j)(z)dz,$$

*and $E_j(\tau)$ is an error term*

$$E_j(\tau) = E_j^{(1)}(\tau)\tau + E_j^{(2)}(\tau)\lambda_j^{-1}\varepsilon_j^a(\tau) + O(\tau^{3/2}),$$

*where*

$$
E_j^{(1)}(\tau) = \left( \mathbb{E}_{\check{\pi}_y^{(j)}} \left[ \frac{x^2}{\mu_j} \right] - \lambda_j^{-1} p_j^{-1} \varepsilon_j^b(\tau)^2 \right) (1 - p_j) + \frac{1}{2} \mathbb{E}_{\check{\pi}_y^{(j)}} \left[ \lambda_j \left( \phi_j'(x)^2 - \phi_j''(x) \right) \right.
$$

$$
\left. - \phi_j'(x)(1 - 2p_j)x \right] - \mathcal{Z}(\varepsilon_j^b(\tau), A_{jj}, y_j)^{-1} \int e^{-\phi_j(z) - \frac{1}{2\sigma^2} [A_{jj}z - y_j]^2} \mathcal{N}(-p_j^{-1}\varepsilon_j^b(\tau), \mu_j)(z)
$$

$$
\times \left[ \frac{1}{2} \left( \lambda_j (\phi'(z)^2 - \phi_j''(z)) - \phi_j'(z)(1 - 2p_j)z \right) + \left( \frac{z + p_j^{-1}\varepsilon_j^b(\tau)}{\mu_j} \right. \right.
$$

$$
\left. \left. + \frac{(z + p_j^{-1}\varepsilon_j^b(\tau))^2}{\mu_j} - \frac{1}{2} \right) (1 - p_j) \right] dz,
$$

*and*

$$
E_j^{(2)}(\tau) = \mathbb{E}_{\check{\pi}_y^{(j)}} \left[ x^2 - \phi_j(x) \right]
$$

$$
- p_j^{-2} \varepsilon_j^b(\tau)^2 - \mathcal{Z}(\varepsilon_j^b(\tau), A_{jj}, y_j)^{-1} \int e^{-\phi_j(z) - \frac{1}{2\sigma^2}[A_{jj}z - y_j]^2} \mathcal{N}(-p_j^{-1}\varepsilon_j^b(\tau), \mu_j)(z)
$$

$$
\times \left[ -\phi_j(z) + z + p_j^{-1}\varepsilon_j^b(\tau) + \left( z + p_j^{-1}\varepsilon_j^b(\tau) \right)^2 - \frac{\mu_j}{2} \right] dz,
$$

*with*

$$
\mathcal{Z}(\varepsilon_j^b(\tau), A_{jj}, y_j) = \int e^{-\phi_j(z) - \frac{1}{2\sigma^2}[A_{jj}z - y_j]^2} \mathcal{N}(-p_j^{-1}\varepsilon_j^b(\tau), \mu_j)(z)dz.
$$

*Proof.* In the following $\mathbb{E}_{\check{\pi}_y^{(j)}}[\psi]$ and $\mathbb{E}_{\check{\pi}_y^{(j)}}[\psi(x)]$ stand for $\int \psi(x) d\check{\pi}_y^{(j)}(x)$. Recall that for $j \leq N$ the $j$-th mode marginal of the approximate posterior distribution $\check{\pi}_y$ is

$$
d\check{\pi}_y^{(j)}(X^{(j)}) = \frac{1}{Z_{\check{\pi}_y^{(j)}}} \exp \left( -\check{\phi}_j(X^{(j)}) - \lambda_j^{-1}\varepsilon_j^a(\tau)\phi_j(X_t^{(j)}) - \frac{1}{2\sigma^2} \left[ A_{jj}X^{(j)} - y_j \right]^2 \right)
$$

$$
\times d\mathcal{N} \left( -\left[ \frac{1}{\check{\mu}_j} + \lambda_j^{-1}\varepsilon_j^a(\tau) \right]^{-1} \lambda_j^{-1}\varepsilon_j^b(\tau), \left[ \frac{1}{\check{\mu}_j} + \lambda_j^{-1}\varepsilon_j^a(\tau) \right]^{-1} \right) (X^{(j)}),
$$

while the true posterior is

$$
d\pi_y^{(j)}(X^{(j)}) = \frac{1}{Z_{\check{\pi}_y^{(j)}}} \exp \left( -\phi_j(X^{(j)}) - \frac{1}{2\sigma^2} \left[ A_{jj}X^{(j)} - y_j \right]^2 \right) d\mathcal{N}(0, \mu_j).
$$

For the unobserved modes $j > N$, we set $A_{jj} = 0$. For each $j$, we have

$$
D_{KL} \left( \check{\pi}_y^{(j)} \,\middle|\middle|\, \pi_y^{(j)} \right)
$$

$$
= \underbrace{\mathbb{E}_{\check{\pi}_y^{(j)}} \left[ \log \mathcal{N} \left( -\left[ \frac{1}{\check{\mu}_j} + \lambda_j^{-1}\varepsilon_j^a(\tau) \right]^{-1} \lambda_j^{-1}\varepsilon_j^b(\tau), \left[ \frac{1}{\check{\mu}_j} + \lambda_j^{-1}\varepsilon_j^a(\tau) \right]^{-1} \right) - \log \mathcal{N}(0, \mu_j) \right]}_{\text{first term}}
$$

$$
+ \underbrace{\mathbb{E}_{\check{\pi}_y^{(j)}} \left( [1 - \lambda_j^{-1}\varepsilon_j^a(\tau)]\phi_j - \check{\phi}_j \right)}_{\text{second term}} + \underbrace{\log \frac{Z_{\pi_y^{(j)}}}{Z_{\check{\pi}_y^{(j)}}}}_{\text{third term}}.
$$

**First term**  We derive

$$\log \mathcal{N}\left(-\left[\frac{1}{\check{\mu}_j}+\lambda_j^{-1}\varepsilon_j^a(\tau)\right]^{-1}\lambda_j^{-1}\varepsilon_j^b(\tau),\left[\frac{1}{\check{\mu}_j}+\lambda_j^{-1}\varepsilon_j^a(\tau)\right]^{-1}\right)(x)$$

$$=-\frac{1}{2}\log(2\pi)-\frac{1}{2}\log\left[\frac{1}{\check{\mu}_j}+\lambda_j^{-1}\varepsilon_j^a(\tau)\right]^{-1}$$

$$-\frac{1}{2}\left[\frac{1}{\check{\mu}_j}+\lambda_j^{-1}\varepsilon_j^a(\tau)\right]\left(x+\left[\frac{1}{\check{\mu}_j}+\lambda_j^{-1}\varepsilon_j^a(\tau)\right]^{-1}\lambda_j^{-1}\varepsilon_j^b(\tau)\right)^2,$$

and

$$\log \mathcal{N}(0,\mu_j)(x)=-\frac{1}{2}\log(2\pi)-\frac{1}{2}\log\mu_j-\frac{1}{2}\frac{x^2}{\mu_j}.$$

Hence

$$\mathbb{E}_{\tilde{\pi}_y^{(j)}}\left[\log\mathcal{N}\left(-\left[\frac{1}{\check{\mu}_j}+\lambda_j^{-1}\varepsilon_j^a(\tau)\right]^{-1}\lambda_j^{-1}\varepsilon_j^b(\tau),\left[\frac{1}{\check{\mu}_j}+\lambda_j^{-1}\varepsilon_j^a(\tau)\right]^{-1}\right)-\log\mathcal{N}(0,\mu_j)\right]$$

$$=\frac{1}{2}\log\left(\frac{1}{e^{-\tau}+(1-e^{-\tau})p_j}+p_j^{-1}\varepsilon_j^a(\tau)\right)-\frac{1}{2}\mathbb{E}_{\tilde{\pi}_y^{(j)}}\left[\frac{1}{\mu_j}\left(\frac{1}{e^{-\tau}+(1-e^{-\tau})p_j}+p_j^{-1}\varepsilon_j^a(\tau)\right)\right.$$

$$\times\left.\left(x+\left[\frac{1}{e^{-\tau}+(1-e^{-\tau})p_j}+p_j^{-1}\varepsilon_j^a(\tau)\right]^{-1}p_j^{-1}\varepsilon_j^b(\tau)\right)^2-\frac{x^2}{\mu_j}\right].$$

We have

$$\frac{1}{2}\log\left(\frac{1}{e^{-\tau}+(1-e^{-\tau})p_j}+p_j^{-1}\varepsilon_j^a(\tau)\right)=\frac{1}{2}\left((1-p_j)\tau+p_j^{-1}\varepsilon_j^a(\tau)\right)+O(\tau^2),$$

and

$$\mathbb{E}_{\tilde{\pi}_y^{(j)}}\left[\frac{1}{\mu_j}\left(\frac{1}{e^{-\tau}+(1-e^{-\tau})p_j}+p_j^{-1}\varepsilon_j^a(\tau)\right)\right.$$

$$\times\left.\left(x+\left[\frac{1}{e^{-\tau}+(1-e^{-\tau})p_j}+p_j^{-1}\varepsilon_j^a(\tau)\right]^{-1}p_j^{-1}\varepsilon_j^b(\tau)\right)^2-\frac{x^2}{\mu_j}\right]$$

$$=\left[(1-p_j)\tau+p_j^{-1}\varepsilon_j^a(\tau)\right]\mathbb{E}_{\tilde{\pi}_y^{(j)}}\left[\frac{x^2}{\mu_j}\right]+2\lambda_j^{-1}\varepsilon_j^b(\tau)\mathbb{E}_{\tilde{\pi}_y^{(j)}}[x]+\lambda_j^{-1}p_j^{-1}\varepsilon_j^b(\tau)^2$$

$$\times\left[1-(1-p_j)\tau-p_j^{-1}\varepsilon_j^a(\tau)\right]+O(\tau^2).$$

**Second term**  We have

$$\mathbb{E}\left[\exp\left(-\phi_j(\widetilde{X}_0)\right)\mid\widetilde{X}_\tau=x\right]=\int\exp(-\phi_j(z))\frac{1}{\sqrt{2\pi v_\tau}}\exp\left(-\frac{(z-m_\tau(x))^2}{2v_\tau}\right)dz,$$

where

$$m_\tau(x)=\frac{e^{-\tau/2}\mu_j}{e^{-\tau}\mu_j+(1-e^{-\tau})\lambda_j}x=\left[1+\left(\frac{1}{2}-p_j\right)\tau+\left(\frac{p_j}{2}-\frac{1}{8}\right)\tau^2\right]x+O(\tau^3),$$

and

$$\sqrt{v_\tau}=\sqrt{\mu_j-\frac{e^{-\tau}\mu_j^2}{e^{-\tau}\mu_j+(1-e^{-\tau})\lambda_j}}=\sqrt{\lambda_j\tau}\left(1-\frac{\tau}{4}-\frac{\tau^2}{32}+O(\tau^{5/2})\right).$$

By the change of variable $w = \frac{z - m_\tau(x)}{\sqrt{v_\tau}}$, we obtain

$$\mathbb{E}\left[\exp\left(-\phi_j(\widetilde{X}_0)\right) \mid \widetilde{X}_\tau = x\right]$$

$$= \int \exp(-\phi_j(z)) \frac{1}{\sqrt{2\pi v_\tau}} \exp\left(-\frac{(z - m_\tau(x))^2}{2v_\tau}\right) dz$$

$$= \int \exp(-\phi_j\left(\sqrt{v_\tau}w + m_\tau(x)\right)) \frac{1}{\sqrt{2\pi}} \exp(-w^2/2) dw$$

$$= \exp(-\phi_j(x)) \left[1 + \left\{\lambda_j\left[\phi_j'(x)^2 - \phi_j''(x)\right] - \phi_j'(x)(1 - 2p_j)x\right\}\frac{\tau}{2} + O(\tau^{3/2})\right]$$

where we used the Taylor expansion for $\exp(-\phi_j\left(\sqrt{v_\tau}w + m_\tau(x)\right))$ as $\sqrt{\tau} \to 0$ and

$$\int \frac{w}{\sqrt{2\pi}} \exp(-w^2/2) = 0, \qquad \int \frac{w^2}{\sqrt{2\pi}} \exp(-w^2/2) = 1.$$

Hence

$$\mathbb{E}_{\tilde{\pi}_y^{(j)}}(\phi_j(x) - \check{\phi}_j(x))$$

$$= \mathbb{E}_{\tilde{\pi}_y^{(j)}} \log\left[1 + \left\{\lambda_j\left[\phi_j'(x)^2 - \phi_j''(x)\right] - \phi_j'(x)(1 - 2p_j)x\right\}\frac{\tau}{2} + O(\tau^{3/2})\right].$$

**Third term**  For the unobserved modes $j > N$, we analyze

$$\frac{\int e^{-\phi_j(z)}\mathcal{N}(0, \mu_j)(z)dz}{\int e^{-\check{\phi}_j(z) - \lambda_j^{-1}\varepsilon_j^a(\tau)\phi_j(z)}\mathcal{N}(-[\check{\mu}_j^{-1} + \lambda_j^{-1}\varepsilon_j^a(\tau)]^{-1}\lambda_j^{-1}\varepsilon_j^b(\tau), [\check{\mu}_j^{-1} + \lambda_j^{-1}\varepsilon_j^a(\tau)]^{-1})(z)dz}. \tag{27}$$

We use that

$$\check{\phi}_j(z) = \phi_j(z) - \left[\lambda_j(\phi_j'(z)^2 - \phi_j''(z)) - \phi_j'(z)(1 - 2p_j)z\right]\frac{\tau}{2} + O(\tau^{3/2}),$$

which implies

$$e^{-\check{\phi}_j(z) - \lambda_j^{-1}\varepsilon_j^a(\tau)\phi_j(z)}$$

$$= e^{-\phi_j(z)}\left(1 + \left[\lambda_j(\phi_j'(z)^2 - \phi_j''(z)) - \phi_j'(z)(1 - 2p_j)z\right]\frac{\tau}{2} - \lambda_j^{-1}\varepsilon_j^a\phi_j(z) + O(\tau^{3/2})\right).$$

Now we consider the density of $\mathcal{N}\left(-[\check{\mu}_j^{-1} + \lambda_j^{-1}\varepsilon_j^a(\tau)]^{-1}\lambda_j^{-1}\varepsilon_j^b(\tau), [\check{\mu}_j^{-1} + \lambda_j^{-1}\varepsilon_j^a(\tau)]^{-1}\right)$:

$$\frac{1}{\sqrt{2\pi[\check{\mu}_j^{-1} + \lambda_j^{-1}\varepsilon_j^a(\tau)]^{-1}}} \exp\left(-\frac{\left(z + [\check{\mu}_j^{-1} + \lambda_j^{-1}\varepsilon_j^a(\tau)]^{-1}\lambda_j^{-1}\varepsilon_j^b(\tau)\right)^2}{2[\check{\mu}_j^{-1} + \lambda_j^{-1}\varepsilon_j^a(\tau)]^{-1}}\right). \tag{28}$$

We have

$$\frac{1}{\sqrt{2\pi[\check{\mu}_j^{-1} + \lambda_j^{-1}\varepsilon_j^a(\tau)]^{-1}}} = \frac{1}{\sqrt{2\pi\mu_j}}\left[1 - \frac{1}{2}(1 - p_j)\tau - \frac{1}{2}p_j^{-1}\varepsilon_j^a(\tau) + O(\tau^2)\right]. \tag{29}$$

We now look at the exponent of (28). Its numerator reduces to

$$(z + p_j^{-1}\varepsilon_j^b(\tau))^2 - 2(z + p_j^{-1}\varepsilon_j^b(\tau))((1 - p_j)\tau + p_j^{-1}\varepsilon_j^a(\tau)) + O(\tau^2),$$

while the reciprocal of its denominator (28) reduces to

$$\frac{1}{2\mu_j}\left[1 + (1 - p_j)\tau + p_j^{-1}\varepsilon_j^a(\tau) + O(\tau^2)\right].$$

Then the exponent of (28) can be expanded as

$$-\frac{1}{2\mu_j}z^2 - \frac{1}{2\mu_j}p_j^{-2}\varepsilon_j^b(\tau)^2 - z\lambda_j^{-1}\varepsilon_j^b(\tau)$$

$$+ \left[\frac{(z + p_j^{-1}\varepsilon_j^b(\tau))}{\mu_j} + \frac{(z + p_j^{-1}\varepsilon_j^b(\tau))^2}{2\mu_j}\right]((1 - p_j)\tau + p_j^{-1}\varepsilon_j^a(\tau)) + O(\tau^2),$$

and the exponential term in (28) becomes

$$
\exp\left(-\frac{(z + p_j^{-1}\varepsilon_j^b(\tau))^2}{2\mu_j}\right)
$$

$$
\times\left[1 + \left(\frac{z + p_j^{-1}\varepsilon_j^b(\tau)}{\mu_j} + \frac{(z + p_j^{-1}\varepsilon_j^b(\tau))^2}{2\mu_j}\right)((1 - p_j)\tau + p_j^{-1}\varepsilon_j^a(\tau)) + O(\tau^2)\right]. \tag{30}
$$

Putting together (29) and (30) we get that the Gaussian density (28) is expanded as

$$
\mathcal{N}(-p_j^{-1}\varepsilon_j^b(\tau), \mu_j)(z)
$$

$$
\times\left[1 + \left(\frac{z + p_j^{-1}\varepsilon_j^b(\tau)}{\mu_j} + \frac{(z + p_j^{-1}\varepsilon_j^b(\tau))^2}{\mu_j} - \frac{1}{2}\right)((1 - p_j)\tau + p_j^{-1}\varepsilon_j^a(\tau)) + O(\tau^2)\right].
$$

We can now expand for small $\tau$

$$
\left[\int e^{-\check\phi_j - \lambda_j^{-1}\varepsilon_j^a(\tau)\phi_j}d\mathcal{N}(-[\check\mu_j^{-1} + \lambda_j^{-1}\varepsilon_j^a(\tau)]^{-1}\lambda_j^{-1}\varepsilon_j^b(\tau), [\check\mu_j^{-1} + \lambda_j^{-1}\varepsilon_j^a(\tau)]^{-1})\right]^{-1}.
$$

Let

$$
\mathcal{Z}_j(\varepsilon_j^b(\tau)) = \int e^{-\phi_j(z)}\mathcal{N}(-p_j^{-1}\varepsilon_j^b(\tau), \mu_j)(z)dz.
$$

We derive

$$
\mathcal{Z}_j(\varepsilon_j^b(\tau))^{-1}\Bigg\{1 - \mathcal{Z}_j(\varepsilon_j^b(\tau))^{-1}
$$

$$
\times \int e^{-\phi_j(z)}\mathcal{N}(-p_j^{-1}\varepsilon_j^b(\tau), \mu_j)(z)\Bigg[\left(\lambda_j(\phi'(z)^2 - \phi_j''(z)) - \phi_j'(z)(1 - 2p_j)z\right)\frac{\tau}{2}
$$

$$
- \lambda_j^{-1}\varepsilon_j^a(\tau)\phi_j(z) + \left(\frac{z + p_j^{-1}\varepsilon_j^b(\tau)}{\mu_j} + \frac{(z + p_j^{-1}\varepsilon_j^b(\tau))^2}{\mu_j} - \frac{1}{2}\right)\left((1 - p_j)\tau + p_j^{-1}\varepsilon_j^a(\tau)\right)\Bigg]dz\Bigg\}
$$

$$
+ O(\tau^{3/2}).
$$

Then (27) can be expanded as

$$
\log\mathcal{Z}_j(0) - \log\mathcal{Z}_j(\varepsilon_j^b(\tau))
$$

$$
- \mathcal{Z}_j(\varepsilon_j^b(\tau))^{-1}\int e^{-\phi_j(z)}\mathcal{N}(-p_j^{-1}\varepsilon_j^b(\tau), \mu_j)(z)\Bigg[\left(\lambda_j(\phi'(z)^2 - \phi_j''(z))\right.
$$

$$
- \phi_j'(z)(1 - 2p_j)z\bigg)\frac{\tau}{2} - \lambda_j^{-1}\varepsilon_j^a(\tau)\phi_j(z) + \left(\frac{z + p_j^{-1}\varepsilon_j^b(\tau)}{\mu_j} + \frac{(z + p_j^{-1}\varepsilon_j^b(\tau))^2}{\mu_j} - \frac{1}{2}\right)
$$

$$
\times ((1 + p_j)\tau + p_j^{-1}\varepsilon_j^a(\tau))\Bigg]dz + O(\tau^{3/2}).
$$

Now let

$$
\mathcal{Z}_j(\varepsilon_j^b(\tau), A_{jj}, y_j) = \int e^{-\phi_j(z) - \frac{1}{2\sigma^2}[A_{jj}z - y_j]^2}\mathcal{N}(-p_j^{-1}\varepsilon_j^b(\tau), \mu_j)(z)dz.
$$

For the observed modes $j \le N$, we get

$$
\log\mathcal{Z}_j(0, A_{jj}, y_j) - \log\mathcal{Z}_j(\varepsilon_j^b(\tau), A_{jj}, y_j)
$$

$$
- \mathcal{Z}_j(\varepsilon_j^b(\tau), A_{jj}, y_j)^{-1}\int e^{-\phi_j(z) - \frac{1}{2\sigma^2}[A_{jj}z - y_j]^2}\mathcal{N}(-p_j^{-1}\varepsilon_j^b(\tau), \mu_j)(z)\Bigg[\left(\lambda_j(\phi'(z)^2 - \phi_j''(z))\right.
$$

$$
- \phi_j'(z)(1 - 2p_j)z\bigg)\frac{\tau}{2} - \lambda_j^{-1}\varepsilon_j^a(\tau)\phi_j(z) + \left(\frac{z + p_j^{-1}\varepsilon_j^b(\tau)}{\mu_j} + \frac{(z + p_j^{-1}\varepsilon_j^b(\tau))^2}{\mu_j} - \frac{1}{2}\right)
$$

$$
\times ((1 - p_j)\tau + p_j^{-1}\varepsilon_j^a(\tau))\Bigg]dz + O(\tau^{3/2}).
$$

$\square$

**Remark C.1.** *If $\phi_j$ is smooth and $\varepsilon_j^b(\tau) = O(1)$, then $E_j(\tau) \to 0$ as $\tau \to 0$.*

# D    Illustrations: Additional Details

Here we provide additional details on the theoretical setup underlying the illustrations. All illustrations were generated on Google Colab (13 GB of RAM), and all code executions took less than one minute[1].

## D.1    Recovering the KL coefficients of the Brownian sheet

The Brownian sheet $B(x_1, x_2)$ is a Gaussian process with zero mean and covariance

$$\text{Cov}(B(x_1, x_2), B(y_1, y_2)) = \min(x_1, y_1) \min(x_2, y_2).$$

Its Karhunen-Loève expansion [50] is

$$B(x_1, x_2) = \sum_{j,k} \phi_{j,k}(x_1, x_2) \eta_{j,k}, \qquad (x_1, x_2) \in [0, 1]^2,$$

where $\eta_{j,k} \sim \mathcal{N}(0, \mu_{j,k})$ are independent Gaussian random variables, and

$$\phi_{j,k}(x_1, x_2) = 2 \sin\left(\pi\left(j - \frac{1}{2}\right) x_1\right) \sin\left(\pi\left(k - \frac{1}{2}\right) x_2\right),$$

$$\mu_{j,k} = \left(\left(j - \frac{1}{2}\right)\pi\left(k - \frac{1}{2}\right)\pi\right)^{-2}.$$

In Section 6, we truncate the KL expansion after $N$ modes

$$B^N(x_1, x_2) = \sum_{j,k=1}^{N} \phi_{j,k}(x_1, x_2) \eta_{j,k},$$

and consider the inverse problem of recovering the first $N^2$ coefficients from noisy observations corresponding to the first $M^2 \leq N^2$ modes

$$y_{j,k} = \tilde{\eta}_{j,k} + \varepsilon_{j,k}, \qquad j, k \leq M,$$

where the prior is $\tilde{\eta}_{j,k} \sim \mathcal{N}(0, \mu_{j,k})$ and the noise $\varepsilon_{j,k} \sim \mathcal{N}(0, \sigma^2)$. This setup satisfies the assumptions of our theory, since the prior diagonal in the KL basis $(\phi_{j,k})$ and the forward map is simply the projection onto these modes, so that

$$A_{j,k,j',k'} = \delta_{j,j'} \delta_{k,k'}, \qquad j, j', k', k \leq M,$$

and zero otherwise. As a result, the posterior for each coefficient remains Gaussian

$$\tilde{\eta}_{j,k} \mid y_{j,k} \sim \pi_{y_{j,k}}^{(j,k)} = \mathcal{N}(m_{j,k}, v_{j,k}),$$

with, for $j, k \leq M$,

$$v_{j,k} = \left(\mu_{j,k}^{-1} + \sigma^{-2}\right)^{-1} = \frac{\mu_{j,k} \sigma^2}{\mu_{j,k} + \sigma^2}, \qquad m_{j,k} = \frac{\mu_{j,k}}{\mu_{j,k} + \sigma^2} y_{j,k},$$

and for $j > M$ or $k > M$ (unobserved modes) the posterior simply coincides with the prior, $v_{j,k} = \mu_{j,k}, m_{j,k} = 0$.

**Experimental details**    In Figure 2, within the theoretical setup described above, we set the noise level $\sigma = 10^{-2}$, chose $N = 200$, and varied the number of observed modes $M^2 = 75^2, 200^2$ to illustrate the discretization-invariance of the preconditioned Langevin sampler. This is confirmed by the small errors reported in the fourth column of Figure 2. The preconditioned Langevin dynamics, using the preconditioner $C_M = \text{Diag}_{1 \leq j,k \leq M}(\lambda_{j,k})$, $\lambda_{j,k} = [\mu_{j,k}^{-1} + \sigma^{-2}]^{-1}$, was run for $5 \cdot 10^3$ iterations with a fixed step-size of $5 \cdot 10^{-1}$. We assumed access to the exact score function, i.e., $\phi = \tau = 0$ in (6).

---

[1]Code to reproduce results can be found at https://github.com/balorenz1/SGM-Inf-Langevin

## D.2 Inverse source problem for the heat equation

Let $\Omega = [0,1]^2 \subseteq \mathbb{R}^2$. Consider $u : \Omega \times [0,T] \to \mathbb{R}$ solving the heat equation

$$\begin{cases} \partial_t u(x,t) = \Delta u(x,t), & (x,t) \in \Omega \times (0,T], \\ u(x,0) = g(x), & x \in \Omega, \\ u(x,t) = 0, & x \in \partial\Omega \times (0,T]. \end{cases}$$

Set $u(x_1, x_2; t) = \sum_{j,k=1}^{\infty} u_{j,k}(t)\,\psi_{j,k}(x_1,x_2)$, where $(\psi_{j,k}, \zeta_{j,k})$ are the Dirichlet eigenpairs of $-\Delta$ on $[0,1]^2$:

$$\begin{cases} -\Delta\psi_{j,k}(x_1,x_2) = \zeta_{j,k}\,\psi_{j,k}(x_1,x_2), & (x_1,x_2) \in [0,1]^2, \\ \psi_{j,k}|_{\partial[0,1]^2} = 0. \end{cases}$$

We have

$$\psi_{j,k}(x_1,x_2) = 2\sin(j\pi x_1)\sin(k\pi x_2), \qquad \zeta_{j,k} = \pi^2(j^2 + k^2).$$

The coefficients evolve as

$$u_{j,k}(t) = e^{-\zeta_{j,k}t}\,g_{j,k}, \qquad g_{j,k} = \langle g, \phi_{j,k}\rangle.$$

In Section 6, we consider the the so-called backward heat equation—the ill-posed inverse problem of recovering the initial condition $g$ from noisy measurements of $u(\cdot, T)$ inside $\Omega$

$$y_{j,k} = e^{-\zeta_{j,k}T}\,g_{j,k} + \varepsilon_{j,k}, \qquad j,k \le M,$$

by adopting a Bayesian approach [3]. We assume a Gaussian prior $g_{j,k} \sim \mathcal{N}(0, e^{-\beta\zeta_{j,k}})$ and independent Gaussian noise $\varepsilon_{j,k} \sim \mathcal{N}(0, \sigma^2)$. The forward map is diagonal in $(\psi_{j,k})$, with

$$A_{j,k,j',k'} = e^{-\zeta_{j,k}T}\delta_{j,j'}\delta_{k,k'}, \qquad j,j',k,k' \le M.$$

As a result, the posterior for each coefficient remains Gaussian

$$g_{j,k} \mid y_{j,k} \sim \mathcal{N}(m_{j,k}, v_{j,k}),$$

with, for $j,k \le M$,

$$v_{j,k} = \frac{e^{-\beta\zeta_{j,k}}\sigma^2}{e^{-(\beta+2T)\zeta_{j,k}} + \sigma^2}, \qquad m_{j,k} = \frac{\mu_{j,k}}{e^{-(\beta+2T)\zeta_{j,k}} + \sigma^2}e^{-\zeta_{j,k}T}y_{j,k}.$$

For $j > M$ or $k > M$ (unobserved modes), the posterior simply coincides with the prior.

**Experimental details** In Figure 3, within the theoretical setup described above, we fixed the noise level at $\sigma = 5 \cdot 10^{-3}$, chose $M = 15$ (i.e. 225 observed modes), and set $T = 0.1$. We then ran the preconditioned Langevin sampler—with the optimal preconditioner $C$ from Theorem 4.1—using the exact score function perturbed by a relative error $\varepsilon_j^a \sim \mathcal{N}(0, 0.1^2)$, scaled by $\tau = 10^{-3}$ in the top row of Figure 3 and by $10^{-1}$ in its bottom row, and with zero bias (i.e. $\varepsilon_j^b = 0$) to simulate a learned score. This sampler was run for $5 \cdot 10^3$ iterations with a fixed step-size of $10^{-2}$. For comparison, we also executed the vanilla Langevin sampler for $1.5 \cdot 10^4$ iterations with a fixed step-size of $10^{-6}$. To further illustrate the quality of our preconditioned posterior samples, Figure 5 below shows uncertainty quantification for Figure 3. For the first 35 modes, we plot the conditional posterior mean (red), the 95% credible interval (orange shading), and the ground truth (dotted black line).

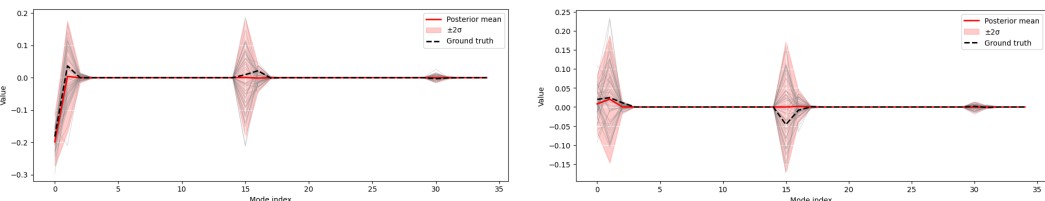

Figure 5: Uncertainty quantification for preconditioned posterior sampling. Left: $\tau = 10^{-3}$. Right: $\tau = 10^{-1}$

