# OpenReview forum: "Preconditioned Langevin Dynamics with Score-based Generative Models for Infinite-Dimensional Linear Bayesian Inverse Problems"
_NeurIPS.cc/2025/Conference — NeurIPS 2025 spotlight_

### Official Review · Reviewer_QcCN · 2025-07-02

**Clarity:** 3
**Significance:** 2
**Originality:** 3
**Rating:** 5
**Confidence:** 4

**Summary:**

The work analyzes Score-based Models for Bayesian inference of linear inverse problems, extending to the case that posterior is supported on an infinite-dimensional space. It is shown that a trace-class preconditioning operator is required for the Langevin diffusion. Sufficient conditions for global convergence in KL divergence are derived and the existence and the form of an optimal preconditioner which depends on the score approximation error and the forward operator are also proved. The analysis covers both Gaussian and non-Gaussian priors.

**Questions:**

1. In Fig 1, the first row of simulation is described as having growing variance. However, it seems to me that the variance stay the same when the mode index is above 60 or so. Can the authors provide further explanation regarding this?

2. Can the author justify the assumption that $A_N$ is non-singular?

3. How plausible are the conditions, $p_j^{-1} \epsilon_j^a(\tau) = O(\tau)$ and $\lambda_j^{-1} \epsilon_j^b(\tau) = O(1)$ in Theorem 3.1? Can the authors provide some insights on these conditions?

4. The assumption that $A_N^\top A_N$ is diagonal in Proposition 4.1 and Theorem 4.1 seems to be rather restrictive.

5. The mean reversion rate $\kappa$ seems to be undefined in Proposition 4.1.

6. Assumption 1 is on the score approximation error, can the authors provide some comments on how this error can be guaranteed to be small as later results depend on this error?

7. In line 280, $|\epsilon^a_j| \ll \lambda_j$ requires approximate error to be small ($\epsilon^a_j$ need to be small) if the scale of the Brownian motion ($\lambda_j$) is small. It will be better if the authors can present some examples for validating such sufficient conditions.

**Ethical Concerns:**

["NO or VERY MINOR ethics concerns only"]

**Final Justification:**

After the rebuttal with the authors and the discussions from the other reviewers, I decided to raise my score from 4 to 5.

My original borderline acceptance rating is because I am not familiar with originality and significance of such work. The detailed discussion has helped me understand the paper better and thus improve my confidence on the paper.

**Limitations:**

Yes.

**Paper Formatting Concerns:**

No major concerns.

**Quality:**

3

**Strengths And Weaknesses:**

Lifting the Bayesian inference algorithms to infinite-dimensional space is well-motivated. The work provides the convergence analysis for both Gaussian and general non-Gaussian priors. It has good amount of originality.

However, the work has some restrictive assumptions and the authors do not justify these well and thus the significance is not obvious to me. For example, how can the non-singularity of $A_N$ and $A_N^\top A_N$ be guaranteed in practical problems? Besides, the sufficient conditions of the optimal preconditioned is not well discussed to provide enough insights.

Overall, it will be better if the authors can provide some examples which satisfies the derived sufficient conditions and thus make the significance of the work more clear and further improve the quality of the work. If the sufficient conditions are easy to validate, then the work will be more of importance.

---

> ### Author Rebuttal · Authors · 2025-07-30
>
> We would like to thank the Reviewer for their constructive feedback.
>
> We are happy to clarify our manuscript in response to the Reviewer's questions. We hope that this could lead to an improvement in their assessment of the paper.
>
> 1. **Figure 1.** We thank the reviewer for spotting this typo. What we intended to say is that the posterior covariance is _not_ trace class; that is, the eigenvalues of its second moment do not decay at infinity---not that the modes exhibit growing variance. As a result, posterior samples do not belong to $H$, which is theoretically inconsistent with the formulation of the problem. Indeed, a Gaussian measure is supported in $H$ only if its second moment is finite on $H$, which is not the case in Figure 1, since the posterior variance is not decaying at fine scales. We will fix this typo in the camera-ready version of the paper.
>
> 2. **Is $A_N$ really non-singular?** We thank the reviewer for pointing out this assumption, which is a leftover from our first  analysis attempts, where we wanted to keep the setting as  simple as possible. In fact, as the reviewer may have noticed, the final analysis presented in the present manuscript does not rely on the non-singularity of $A_N$; nowhere in the main results or proof do we require that $(A_N^\top A_N)\_{jj}\neq0$. The only part that may require a minor adjustment is lines 234-235, but this does not pose any issue: if $(A_N^\top A_N)\_{jj}=0$, then the convergence rate for that mode is simply $\lambda\_j/\mu\_j$. We will remove this assumption in the camera-ready version of the paper.
>
> 3. **Assuming $A^\top A$ is diagonalizable.** This is a common assumption in many linear Bayesian inverse problem settings and their theoretical analysis; see, for example, the foundational works of Stuart [3] and Knapik et al. [4]. Moreover, in many classical linear inverse problems where $A$ is compact---such as the heat equation, tomography, or inverse scattering for Schr\"odinger-type operators under the Born approximation---one can always find a basis in which $A_N^\top A_N$ is diagonal. That said, the diagonalization assumption is not essential to our analysis; it is adopted primarily to keep the presentation as clean, interpretable, and accessible as possible. For example, the asymptotic expansion in Eq. (10) of Theorem 3.1 can be extended to non-diagonal  $A_N^\top A_N$, after some leg-work, by replacing scalar expansions with the corresponding matrix-series expansions; the part regarding the higher-order modes remains the same. Likewise, in Section 4 one can verify that, under a perfect score function, the optimal preconditioner still takes the form $C=[C_\mu^{-1} + \sigma^{-2} A^\top A]^{-1}$. We will add a remark including these additional considerations.
>
> 4. **Missing definition of the mean reversion rate.** Thank you for noticing this; we will include the definition in the camera-ready version.
>
> 5. **Conditions on score approximation error (answer to questions 3,6,7).** The affine-error decomposition in Assumption 1 is natural and follows from the co-diagonalization assumption for $C$ and $C_\mu$, which is standard in the literature on infinite-dimensional diffusion models [20,48].  In lines 194--199, we provide a qualitative analysis of the impact of $\epsilon^a_j$ and $\epsilon^b_j$ in Proposition 3.1. From the remarks, it is clear that assuming $p_j \epsilon_j^a$ to be small and $\lambda_j^{-1} \epsilon_j^b$ to be bounded is essential for achieving an accurate sampling of the posterior. Theorem 3.1 then makes these observations more quantitative, with assumptions that directly reflect the insights of Proposition 3.1. Since formula (10) is an equality, any significant deviation of $\epsilon^b_j$ and $\epsilon^a_j$ from the stated assumptions would necessarily result in a large sampling error at leading order, preventing an accurate recovery of the posterior mode. In this sense, Theorem 3.1 highlights the interplay between the data spectral properties and the design of the preconditioned Langevin sampler. It provides a sharp characterization of the link between the score approximation  error and the posterior sampling  error, showing that if the score approximation error is too large, it can  significantly impact the overall  posterior sampling error, potentially leading to catastrophic consequences for the error bound. To our knowledge, we are the first ones to provide sufficient conditions that ensure the boundedness of this error in the setting of infinite dimensional Bayesian linear inverse problems. Whether these conditions on the score approximation error are easy to satisfy in practice is an important question for future research, especially given that there is currently no literature on how to control the error arising from training infinite-dimensional score-based generative models to approximate the true score function, see [20, 42, 43, 45, 48]. With this paper, we hope to have laid down not only the theoretical framework but also the practical motivation for why such an analysis is essential to fully understand the infinite-dimensional setting.  We thank the reviewer for bringing up this point; in the camera-ready version, we will include these considerations in the Discussion and Future Work section, clarify the connection between the discussion in lines 194--199 and the assumptions in Theorem 3.1, and elaborate on the implications of Remark 3.1.

---

> > ### Comment · Reviewer_QcCN · 2025-08-04
> >
> > The authors have addressed my concerns. However, I am still unsure about the significance of such work. I will keep the original score.

---

> > > ### Author Response · Authors · 2025-08-07
> > >
> > > Dear Reviewer QcCN,
> > >
> > > We are happy that we could address your concerns in our rebuttal. Regarding the significance of our contribution, we would like to emphasize that the topic is the subject of ongoing and very recent research efforts -- for instance:
> > >
> > > > Pidstrigach, J., Marzouk, Y., Reich, S., & Wang, S. (2024). Infinite-dimensional diffusion models. Journal of Machine Learning Research, 25(414), 1-52.
> > >
> > > We believe that there remains substantial room for further progress in this area. Our hope is that our contribution will help motivate the line of research dedicated to infinite-dimensional diffusion models: we strongly believe that the "Bayesianize-then-discretize" approach deserves to be highlighted in comparison to traditional "discretize-then-Bayesianize" methods.

---

### Official Review · Reviewer_PRj9 · 2025-07-02

**Clarity:** 3
**Significance:** 3
**Originality:** 3
**Rating:** 5
**Confidence:** 3

**Summary:**

The authors analyze Langevin dynamics driven by learned scores in the infinite-dimensional (function) space setting. In particular, the authors studied how the approximation error of the learned score impacts the stationary distribution.

**Questions:**

- What system knowledge is required to compute the preconditioner and what are the computational costs? For example, it seems a complete diagonalization of the forward operator is needed, which can be costly.

- On line 198 the authors discuss the bias introduced by the learned score. How large is this in the numerical examples? For which types of inverse problems is this a major issue and does it give insights when learned priors are useful and when not?

**Ethical Concerns:**

["NO or VERY MINOR ethics concerns only"]

**Final Justification:**

The authors provided a few more clarifications. I considered this paper as strong before the rebuttal and the answers confirmed me in this rating.

**Limitations:**

Yes

**Quality:**

3

**Strengths And Weaknesses:**

Strengths:

+ Using score-based models for sampling in inverse problems (rather for generating new data) is an important research direction. The authors provide convincing examples why inverse problems coming from PDE settings should be studied in the infinite-dimensional setting, which is largely ignored in the more common "generative modeling" setting.

+ The analysis leads to actionable insights, namely a preconditioner. Numerical results (e.g., Figure 3) demonstrate that using the preconditioner stabilizes the inversion.

+ The authors also treat the case of non-Gaussian priors, as long as they are abs continuous to a Gaussian reference (standard assumption).

Weaknesses:
- The analysis is present only for the case that A^T A is diagonalizable. The authors comment on this limitation and state that more technical argument can handle the more general case but do not provide details.

- The computational costs of obtaining the preconditioner are not discussed, especially what information about the forward model and the learned score are needed.

---

> ### Author Rebuttal · Authors · 2025-07-30
>
> We would like to thank the Reviewer for their positive feedback.
>
> We are happy to clarify our manuscript in response to the Reviewer's remarks and questions.
>
> 1. **Assuming $A^\top A$ is diagonalizable.**  This assumption is common in many linear inverse problem settings and their theoretical analysis [3, 4]. We adopted it because we believe that focusing on the diagonal case provides the clearest exposition, keeping our analysis as clean, interpretable, and insightful as possible. Moreover, for any linear inverse problem where $A$ is compact, one can always find a basis in which $A_N^\top A_N$ becomes diagonal. However, what we wrote in the Discussion and Future Work section remains true---the main takeaway of our analysis holds even without the diagonalization assumption. The asymptotic expansion in Eq. (10) of Theorem 3.1 can be extended to non-diagonal  $A_N^\top A_N$, after some leg-work and under appropriate assumptions on the structure of the score approximation error, by replacing scalar expansions with the corresponding matrix-series expansions; the part concerning the higher-order modes remains unchanged. Similarly, in Section 4 one can verify that, under a perfect score function, the optimal preconditioner still takes the form $C=[C_\mu^{-1} + \sigma^{-2} A^\top A]^{-1}$. We can add a remark including these additional considerations.
>
> 2. **Computing the preconditioner.** Indeed, exactly computing the optimal preconditioner would require the diagonalization of $A_N^\top A_N$, which can be computationally expensive. However, a simple and practical alternative would be to select a preconditioner that is as close as possible to the prior covariance; as shown in the analysis of Theorem 4.1, for the higher-order modes, the leading order-term of the preconditioner coincides with the inverse of the prior covariance. This choice is particularly justified when the prior decays quickly, in which case $\mu_j^{-1}\gg \sigma^{-2} (A^T_N A_N)_{jj}$ for the lower-order modes. Any additional information about the posterior covariance or the
> score approximation error can then be used to refine this initial approximation. We can add a remark
> on this point, although we emphasize that a detailed investigation of the computational cost of
> constructing the optimal preconditioner lies beyond the scope of this paper and is left for future work.
>
> 3. **Bias and implications**. In the numerical examples, we set $\epsilon^b\_j =0$. Since formula (10) is an equality, a large value of  $\epsilon^b\_j$ would necessarily lead to a large sampling error, preventing an accurate recovery of the posterior mode. In fact, learning an incorrect score can significantly impact the posterior sampling---a challenge that already exists in finite dimensions but can become catastrophic when attempting to control the global error in the infinite-dimensional setting. In this sense, our result highlights the interplay between
> the data spectral properties and the design of the preconditioned Langevin sampler; understanding how these spectral properties, together with the structure of the inverse problem, influence the learnability of the score  is an important question for future research, especially given that there is currently
> little literature on how to control the error arising from training infinite-dimensional score-based
> generative models to approximate the true score function (see [20, 42, 43, 45, 48]). Our contribution here is a first step toward a theoretical understanding of this issue,
>    in a setting that allows for rigorous  analysis of when a learned prior can provide useful insight.

---

> > ### Comment · Reviewer_PRj9 · 2025-08-03
> >
> > I thank the reviewers for the answers and hope they will incorporate these clarifications in the final version. I maintain my Accept rating.

---

### Official Review · Reviewer_pjRJ · 2025-07-03

**Clarity:** 3
**Significance:** 3
**Originality:** 3
**Rating:** 5
**Confidence:** 2

**Summary:**

This paper studies diffusion modeling in the setting of infinite-dimensional linear inverse problems. While the importance of pre-conditioning has been noted for other solutions of inf-dim inverse problems, it has been absent from SGMs, leading to pathological results when the discretization is refined.

This paper notes the importance of pre-conditioning by starting from a Gaussian setting and establish the existence of an optimal pre-conditioner. The paper then extends this result in the non-Gaussian setting and present similar results. The paper then validates the result from 2 linear inverse problem and illustrates that pre-conditioning is important in robustly recovering the true source of the linear inverse problems.

**Questions:**

- The paper studies a non-Gaussian setting, but the 2 illustrations seem to both stem from Gaussians. Is this an accurate characterization? Is it possible to illustrate a non-Gaussian example?
- Is it possible to obtain the optimal pre-conditioner outside of illustrative examples? If not, does this paper point to a possible solution to the problem of choosing a sensible preconditioner in a practical setting?

**Ethical Concerns:**

["NO or VERY MINOR ethics concerns only"]

**Limitations:**

Yes

**Quality:**

3

**Strengths And Weaknesses:**

I am quite unfamiliar to the field of Bayesian inverse problems, so I find it quite difficult to assess the novelty of this work with little knowledge of related work. Therefore, while I recommend acceptance for this paper, my confidence level is low.
## Strengths
Within the scope of the linear inverse problem, this paper does a good job of explaining and demonstrating the role of the preconditioner.

- The paper is well-written with an easy-to-follow Gaussian example that illustrates to readers who might not be familiar with the linear inverse problem (such as myself) to understand.
- The paper's illustrations seem to support its robustness towards discretization, as well as the failure of vanilla Langevin.
- The idea of an optimal preconditioner seems to be of wider interest

## Weaknesses
- The numerical illustrations are quite difficult to follow. For example, Figure 4 comes with very little explanation and I find it hard to understand the purpose of the figure.
- The illustrations are largely synthetic example that does not include learned score functions, but instead rely on the ground truth score functions with perturbations.
- The assumptions of joint diagonalizability and a diagonal $A^\top A$ seem quite restrictive.

---

> ### Author Rebuttal · Authors · 2025-07-29
>
> We would like to thank the Reviewer for their positive feedback.
>
> We are happy to clarify our manuscript in response to the Reviewer's thoughtful remarks and questions. We hope this could lead to an increase in their rating (or confidence score).
>
> 1. **Explaining Figure 4.** Figure 4 shows the empirical autocorrelation function (ACF) of the first ten modes, comparing preconditioned  versus vanilla Langevin sampling. The $x$-axis is the lag, and the $y$-axis is the autocorrelation, which measures how strongly each mode at step $k$ remains correlated with its initial value. Under preconditioning with the optimal $C$, all the modes ACFs show uniform and rapid decay, meaning that they converge uniformly and quickly to the posterior, as the theoretical analysis of Section 4 anticipated. In contrast, vanilla Langevin displays slow, non-uniform decay. We agree this figure would benefit from a clearer explanation, and we will use the extra page in the camera-ready version to add a more detailed description. We thank the reviewer for the suggestion.
>
> 2. **Synthetic examples.** The synthetic examples are proposed to illustrate the theoretical results, which are the main contributions of the paper. This led us to choose examples that satisfy the key assumptions---namely, the conditions on the score error from Assumption 1, Theorem 3.1 and Theorem 4.1. For this reason, rather than using a learned score model, we simulate an imperfect score by perturbing the analytical score of the prior.  This is a common approach in the literature; see, for example, the extended numerical experiments of [24], which we cite in the Introduction of our paper.
>
> 3. **Assumptions on co-diagonalization of $C$, $C_\mu$ and $A^\top A$.** The co-diagonalizability assumption for $C$ and $C_\mu$ is standard in the literature on infinite-dimensional diffusion models [20, 21, 48]. As for the assumption that $A^\top A$ can be diagonalized, it is common in many linear Bayesian inverse problem settings and their theoretical analysis; see, for example, the foundational works of Stuart [3] and Knapik et al. [4]. Moreover, in many classical linear inverse problem where $A$ is compact---such as the heat equation, tomography, or inverse scattering for Schr\"odinger-type operators under the Born approximation---one can always find a basis in which $A_N^\top A_N$ is diagonal. However, we emphasize that the main conclusions of our analysis remain valid even without the diagonalization assumption. For example, the asymptotic expansion in Eq. (10) of Theorem 3.1 can be extended to non-diagonal  $A_N^\top A_N$, after some leg-work, by replacing scalar expansions with the corresponding matrix-series expansions; the part regarding the higher-order modes remains the same. Likewise, in Section 4 one can verify that, under a perfect score function, the optimal preconditioner still takes the form $C=[C_\mu^{-1} + \sigma^{-2} A^\top A]^{-1}$. We will add a remark including these additional considerations and thank the reviewer for pointing out the need and relevance of these comments.
>
> 4. **Non-Gaussian setting and illustrations.** The paper presents analysis for both Gaussian and non-Gaussian settings. But since the key insights are closely aligned in both cases  (see, for example, lines 194-199; 230-236; 272-281), we preferred focusing on the Gaussian case in the illustrations.
> This allows for explicit expressions for the mean reversion rate, avoiding the added complexity of working with upper and lower bounds of $\partial^2_j \phi_j$, and enabling a cleaner illustration of the main results of the paper. That said, our framework does allow for a non-Gaussian examples; we will explore this if such an example adds further insight into the core theoretical contributions of the paper.
>
> 5. **Choosing a preconditioner in practical settings.** A simple and practical choice is to select a preconditioner that is as close as possible to the prior covariance; as shown in the analysis of Theorem 4.1, for the higher-order modes, the leading order-term of the preconditioner is simply the inverse of the prior covariance. This choice is especially reasonable when the prior decays quickly, in which case $\mu_j^{-1}\gg \sigma^{-2} (A^T_N A_N)_{jj}$for the lower-order modes. Ideally, any available information about the posterior covariance and the score approximation error can be used to refine this first approximation. We can add a remark on this point,  although we emphasize that a detailed investigation of the computational cost of approximating the optimal preconditioner lies beyond the scope of this work.

---

> ### Author Response · Authors · 2025-08-07
>
> Dear Reviewer pjRJ,
>
> We hope this message finds you well. We are very grateful for the time and effort you've dedicated to reviewing our work. We did our best to address your comments thoroughly in our rebuttal, and we hope our responses were clear and helpful.
>
> Please don’t hesitate to let us know if anything remains unclear or if further clarification is needed. We would be happy to elaborate.

---

> > ### Comment · Reviewer_pjRJ · 2025-08-07
> >
> > Thank you for your response and it is helpful in helping me understand the finer details about the paper - I maintain my accept rating and have no further questions at this time.

---

### Official Review · Reviewer_jf1T · 2025-07-03

**Clarity:** 4
**Significance:** 3
**Originality:** 3
**Rating:** 5
**Confidence:** 3

**Summary:**

This paper considers an infinite-dimensional version of the Langevin diffusion sampler to solve Bayesian inverse problems of form $y = AX_0 +n$, for $A$ a linear operator, $X_0\sim \mu$ an unknown Hilbert space valued random variable, and $y, n$ finite-dimensional, with $n$ given as Gaussian observation noise. Under this setup, the aim is to find the posterior of $X_0$ given the observations. In the infinite-dimensional case the diffusion process is driven by a $C$-Wiener process. This $C$ acts as a preconditioner and the aim of the paper is to understand how this preconditioner influences the error of the learned score. To start the analysis, it is assumed that the prior of $X_0$ is Gaussian. Under this assumption the paper provides the stationary distribution of the preconditioned Langevin diffusion in terms of the error of the score approximation as well as the KL-divergence between the true and approximate posteriors. It also considers how to choose an optimal preconditioner $C$. Finally, the paper generalises these results to the case where the prior is absolutely continuous with respect to a Gaussian reference measure.

**Questions:**

See the weaknesses for questions.

**Ethical Concerns:**

["NO or VERY MINOR ethics concerns only"]

**Final Justification:**

The rebuttal addressed my main concern between the similarities between a prior result. The paper is well-written and studies an interesting problem.

**Limitations:**

Yes.

**Paper Formatting Concerns:**

None.

**Quality:**

3

**Strengths And Weaknesses:**

Strengths:

1. The submission is well-written and structured, making it easy to follow. Moreover, the mathematics is well presented with all assumptions clearly stated.

2. The analysis provides insights into the optimal preconditioner $C$ in the case of Langevin samplers which I think is a valuable result. It is also shown how the preconditioner influences the convergence.

3. I enjoyed the remarks and comments surrounding the statements of the main theorems and propositions. I found them useful for interpreting the results.


Weaknesses:

1. Pidstrigach et al. [48] also provide a similar result about the optimal $C$ based on the Wasserstein-2 distance between the true data distribution $\mu$ and the learned sample distribution (see Theorem 14 and Section 6). Although the problem setting is a bit different, in that they are studying time-reversals rather than Langevin-based samplers for inverse problems, the results seem very related. Could you provide some more details on the similarities and differences to this result?

2. It is assumed that the data distribution is absolutely continuous with respect to a Gaussian measure. Could you shed some light on how restrictive this assumption is in practice? For example, by providing some examples of applications where this would hold.

3. In the appendix, Theorem 3.1. is also generalised to the non-Gaussian case, however I don’t think this result is mentioned in the main paper. I think it would be nice to include this somewhere; if not the whole statement then perhaps a sentence linking to the result in the appendix.

4. The experiments provided are quite toy, consisting of experiments on Brownian sheets and the heat equation. However, I only see this as a minor weakness, since the main objective of the paper is theoretical and in my view the experiments serve more as an illustration.

5. There’s a small mistake in Line 318, which currently reads “…about nonlinear inverse problem?”. This is minor, and does not affect my judgment of the paper.


In summary, I think this is a nice paper with interesting results and I recommend acceptance. However, I’m currently a little unsure on how different the results are to that in Theorem 14 of Pidstrigach et al. If the authors could make this clearer I would be willing to improve my score.

---

> ### Author Rebuttal · Authors · 2025-07-29
>
> We thank the Reviewer for their positive feedback and thoughtful questions. We are happy to clarify their doubts and hope this could lead to an increase in their score.
>
> 1. **Similarities and differences with Theorem 14 of [48].** We thank the reviewer for drawing our attention to this specific result in the foundational work by Pidstrigach et al. We are happy to elaborate on the differences and similarities between the two results, and we plan to include this discussion in the camera-ready version of the paper.
> As the reviewer correctly notes, the settings are different: Pidstrigach et al. study the problem of sampling an unknown data distribution using a score learned from samples of that distribution, whereas we use a score-based generative model trained to learn the prior in order to sample the posterior.
> Setting these differences aside, and assuming in Pidstrigach et al. that
> $$
> \mu_{\text{data}}(dx) \propto \exp(-\frac{1}{2\sigma^2}\|Ax- y\|^2)  \mathcal{N}(0,C_\mu)(dx),
> $$
> with $C_\mu= \text{Diag }(\mu\_j)$, and, for simplicity, $A = \text{Diag}({A}\_{jj})$, one can show that the squared Wasserstein distance between $\mu_{\text{data}}$ and $\mathcal{N}(0,C)$, with $C=\text{Diag}(\lambda_j)$, satisfies
> $$
> W^2_2(\mu_{\text{data}}, \mathcal{N}(0,C))=\sum_{j=1}^\infty \left( \lambda_j^{1/2} - [\sigma^{-2}A^2_{jj} + \mu_{j}^{-1}]^{-1/2} \right)^2 + \text{const.}$$
> It follows that the optimal $C$ minimizing the upper bound in Eq. (14) of Pidstrigach et al. paper is given by
> $\lambda_j = [\sigma^{-2}A^2_{jj} + \mu_{j}^{-1}]^{-1}$, which coincides with the covariance of the target data distribution---as expected. In particular, under the assumption of a perfect (i.e. error-free) score, our framework leads to the same optimal covariance. This result can be extended to non-Gaussian priors that are absolutely continuous with respect to a Gaussian reference measure ${\cal N}(0,C_\mu)$,
> by following the arguments developed after Proposition 5.2 in lines 275-280.
> However, we stress that this result does not follow in our case from minimizing an upper bound on the convergence error, as in Theorem 14 of Pidstrigach et al.
> In this sense, our analysis of preconditioning runs in parallel, with differences arising from the distinct objectives of each work.  For example, our preconditioner is derived explicitly; it follows directly from the mean reversion rate of the Langevin dynamics, meaning that the optimal covariance $C$ we find is not merely minimizing an upper bound---it is, under our assumptions, the best achievable in practice, for the purpose we have in mind (to ensure uniform rate of convergence across all modes). Moreover, our optimal $C$ explicitly incorporates the effect of score estimation errors, whereas theirs does not. We believe that these differences do not reflect a limitation---or superiority---of one work over the other, but rather the distinct aims of the two works: Pidstrigach et al. is bounding the convergence error of the reverse SDE dynamics, while our goal is to ensure uniform convergence across all modes in Langevin sampling.
> In this light, the closeness of the two results is particularly interesting. This comparison adds valuable context to our work and helps connect different lines of research on infinite-dimensional score-based generative  modeling and we think reinforces the relevance of both results. We are grateful to the reviewer for pointing this out and will make sure to include the discussion as a remark in the final version of the paper.
>
> 2. **Assuming a prior absolutely continuous with respect to a Gaussian measure.** This is a standard assumption in infinite-dimensional PDE-based Bayesian inverse problem theory (see the foundational works of Stuart [3], Knapik et al. [4], and collaborators [a--e]). One typically begins with a Gaussian prior on the unknown field (e.g. Mat\'ern or squared-exponential covariance), and, via the Bayes' theorem, obtains that the posterior measure is absolutely continuous with respect to the Gaussian prior---a crucial property throughout both theory and applications. Gaussian priors are a popular choice because they can be conveniently defined via covariance operators given by integer or fractional inverse powers of differential operators, are relatively straightforward to sample from, and lead to well--understood H\"older and Sobolev regularity of their realizations, which are typically encoded through assumptions on the decay rate of the eigenvalues of the prior covariance operator.
>  Assuming absolute continuity with respect to a Gaussian reference measure preserves these key properties---such as the connection between covariance structure and regularity---while allowing mild non-Gaussian modification (e.g. spatially varying penalties, bounded exponential tilts that induce skewness or sparsity), which have been shown useful in applications like biology [f] and perimeter learning [g]. In addition to their modelling flexibility, such priors offer a structural advantage: they can be regarded as conjugate priors for the inverse problems. That is, the posterior distributions associated to these priors have the same form---they are absolutely continuous with respect to the same reference Gaussian measure. This invariance makes it possible to study how a likelihood function updates a prior distribution, and also to implement sequential strategies in which the posterior from one step naturally serves as the prior for the next.
>
> 3. **Include generalization of Theorem 3.1 in the main paper**: We thank the reviewer for the suggestion. We fully agree and will include a more direct reference to Theorem C.1 making use of the additional page allowed in the camera-ready version.
>
> 4. **Small mistake in line 318**: Thank you for spotting this typo; we will fix it.
>
>
> **References**
>
> [a] Hairer, Stuart, Voss (2009), Signal Processing Problems on Function Space: Bayesian Formulation, Stochastic PDEs and Effective MCMC Methods
>
> [b] Dashti, Stuart (2011), Uncertainty Quantification and Weak Approximation of an Elliptic Inverse Problem
>
> [c] Beskos, Roberts, Stuart, Voss, (2008) MCMC methods for diffusion bridges
>
> [d] Cotter, Roberts, Stuart, White (2013) MCMC methods for functions: modifying old algorithms to make them faster
>
> [e] Cotter, Dashti, Stuart (2010) Approximation of Bayesian inverse problems for PDEs
>
> [f] Croix, Durrande, Alvarez (2018) Bayesian inversion of a diffusion evolution equation with application to Biology
>
> [g] Dunlop, Elliott, Hoang, Stuart (2017) Reconciling Bayesian and total variation regularization methods for binary inversion

---

> > ### Comment · Reviewer_jf1T · 2025-08-04
> >
> > Thank you for your detailed answers. My main concern previously was in the similarity of the result to Theorem 14 of [48], which has now been addressed and I will increase my score accordingly.

---

### Note · Authors · 2025-08-12

We thank all the reviewers for their positive feedback and suggestions. We take the opportunity offered by the Author Final Remarks to recap our contributions and highlight key points of the Author-Reviewer discussion.

Our work analyzes Langevin dynamics driven by score-based generative models (SGMs) used as priors for infinite-dimensional linear inverse problems. To ensure samples lie in a suitable Hilbert space $H$, they must have finite energy with respect to the norm of $H$, i.e. their covariance must be trace class. Naively applying finite-dimensional sampling procedures can slow convergence or lead to instability when solving the inverse problem. This is why it is crucial to design the algorithm in that setting first, and then apply it to finite-dimensional discretizations.

Our main focus was:

1. Define a method that provably samples a posterior in infinite dimensions, with quantitative analysis giving conditions on the score approximation error such that the KL divergence between the posterior and the sampler's stationary distribution doesn't diverge.
2. Treat rigorously stability issues from the trace class prior, which make the Langevin sampler unstable as the problem discretization is refined, by defining a preconditioner $C$ for which we find the optimal form ensuring uniform convergence across all posterior modes. Numerical results compare our theoretically grounded infinite-dimensional approach to a vanilla finite-dimensional one.

During the discussion phase, we addressed the reviewers main concerns. Some (pjRJ, PRj9, QcCN) asked about the assumption that $C_\mu$, $C$, and $A^T A$ are co-diagonalizable. While co-diagonalization of $C_\mu$ and $C$  is standard [20,48], we note that assuming $A^T A$ is diagonalizable covers a wide range of problems and is mainly used to simplify the analysis; our main conclusions hold without it. On preconditioner choice with minimal prior knowledge (pjRJ,PRj9), our theory suggests starting with one close to the prior covariance, and that any available information about the posterior covariance or the score approximation error can be used to refine it.  On related work (jf1T), we connect to [48], who studied the spectral properties of data in infinite-dimensional SGMs in a different setting and with different objectives. We note that the two contributions are complementary and eventually recommend the same types of preconditioner. We also thank QcCN for noting that assuming $A_N$ invertible is unnecessary.

---

### Decision · Program_Chairs · 2025-09-17

**Decision:**

Accept (spotlight)

**Comment:**

This manuscript provides a theoretical study of an infinite-dimensional version of the Langevin diffusion sampler. All reviewers agree that the derived results present a valuable and interesting contribution to the field (clearly formulated and well-presented). During the rebuttal phase, the authors could largely clarify all concerns raised during the initial review (also leading to score increases), including how the current results are novel compared to prior work. From my perspective, this is already a solid paper and if all requested changes are included in the final version, this presents a solid contribution to NeurIPS. Hence, I am recommending acceptance.